# Quantifying the adaptive landscape of commensal gut bacteria using high-resolution lineage tracking

Daniel P. G. H. Wong[1] & Benjamin H. Good [1,2,3] ✉

Gut microbiota can adapt to their host environment by rapidly acquiring new mutations. However, the dynamics of this process are difficult to characterize in dominant gut species in their complex in vivo environment. Here we show that the fine-scale dynamics of genome-wide transposon libraries can enable quantitative inferences of these in vivo evolutionary forces. By analyzing >400,000 lineages across four human *Bacteroides* strains in gnotobiotic mice, we observed positive selection on thousands of cryptic variants − most of which were unrelated to their original gene knockouts. The spectrum of fitness benefits varied between species, and displayed diverse tradeoffs over time and in different dietary conditions, enabling inferences of their underlying function. These results suggest that within-host adaptations arise from an intense competition between numerous contending variants, which can strongly influence their emergent evolutionary tradeoffs.

The mammalian gut is home to a diverse microbial community comprising hundreds of coexisting strains. High rates of turnover endow these communities with a capacity for rapid evolutionary change. Time-resolved sequencing has started to illuminate this process, with several recent studies in mice[1–7] and humans[8–14] documenting genetic variants sweeping through local populations of gut bacteria on timescales of weeks and months. This strain-level variation can alter metabolic phenotypes[2,15–17], influencing the breakdown of drugs[18] and the invasion of external strains[4,19]. Yet despite their importance, the evolutionary drivers of this in vivo adaptation—and their dependence on the host environment—are only starting to be uncovered.

Traditional sequencing approaches have a limited ability to address these questions since they can only observe the handful of lineages that manage to reach appreciable frequencies within a host. By this time, successful lineages have often acquired multiple distinct mutations[4,8,14]. This makes it difficult to resolve their underlying fitness benefits or the pleiotropic tradeoffs that they encounter in different host conditions[2]. It also prevents us from observing the other contending mutations that—through a combination of luck and merit—were outcompeted before they were able to reach appreciable frequencies within their host.

Barcoded lineage tracking provides a powerful alternative, enabling quantitative fitness measurements of thousands of independent mutations within a single population[20]. However, existing methods for high-throughput isogenic barcoding require specialized genetic tools and have previously been limited to laboratory strains of yeast[20,21] and *E. coli*[22]. Here we show that similar evolutionary inferences can be obtained from genome-wide transposon insertion sequencing (Tn-Seq) libraries[23–27], which are routinely employed in functional genomics settings. Tn-Seq libraries are traditionally used to identify conditionally essential genes in various bacterial species and environments[23–25,28], including several recent in vivo studies in gnotobiotic mice[17,26,29,30]. We aimed to exploit this same technique as a crude form of genetic barcoding, by focusing on the vast majority of Tn insertions that fall in genes without obvious growth defects. We reasoned that the fine-scale dynamics of these lineages could provide a scalable approach for measuring in vivo evolutionary forces in complex communities like the gut microbiota.

Here we illustrate this approach by analyzing the dynamics of >400,000 lineages from a previous transposon screen of four human *Bacteroides* strains in gnotobiotic mice. We uncover evidence for positive selection on thousands of cryptic variants—most of which

[1]Department of Applied Physics, Stanford University, Stanford, CA 94305, USA. [2]Department of Biology, Stanford University, Stanford, CA 94305, USA. [3]Chan Zuckerberg Biohub–San Francisco, San Francisco, CA 94158, USA. ✉e-mail: bhgood@stanford.edu

are unrelated to their original gene knockouts. We develop a mathematical framework for quantifying this adaptive landscape, and how it varies across different bacterial species, hosts, and dietary conditions. We show how the statistical features of this landscape can shed light on the functional targets of within-host adaptation and the evolutionary tradeoffs that emerge among successful lineages. Together, these results demonstrate that lineage tracking can be a powerful tool for resolving the dynamics of within-host evolution.

## Results

### Fine-scale dynamics of Tn-Seq lineages reveal rapid in vivo evolution

To illustrate our approach, we reanalyzed data from a previous transposon screen of multiple commensal gut bacteria in gnotobiotic mice[29]. Tn-Seq libraries of four human *Bacteroides* strains—*B. cellulosilyticus* (*Bc*), *B. ovatus* (*Bo*), and two strains of *B. thetaiotaomicron* (*Bt*-VPI and *Bt*-7330)—were combined with 11 other species and gavaged into 20 individually caged mice. Mice were maintained on either a low-fat/high-plant polysaccharide diet (LF/HPP), a high-fat/high-sugar diet (HF/HS), or alternating sequences of the two (HLH/LHL) for 16 days, with Tn-Seq measurements performed on fecal samples collected at three timepoints (Fig. 1a). In their original study, Wu et al.[29] used these data to show that ~10–30% of gene knockouts displayed a consistent fitness cost in at least one of the diets during the first 16 days of colonization. After excluding the Tn insertions in these and other "fitness determinant" genes, we identified a collection of ~60,000–150,000 mutants in each library that were suitable for high-resolution lineage tracking (Fig. 1b; see the "Methods" section). By monitoring the relative frequencies of these Tn lineages over time, we

sought to quantify the additional evolutionary forces that acted within these populations during the first two weeks of colonization.

Consistent with previous observations in other bacterial species[1–4], we found that a handful of lineages expanded to intermediate frequencies (>1%) in vivo by day 16 (Fig. 1c–f), indicating rapid positive selection on a subset of the lineages. In most of these cases, we observed that the other Tn insertions in the same genes declined over the same time interval (Fig. 1g and Supplementary Fig. 1). This suggests that the fitness benefits of the expanding lineages did not derive from their original Tn insertions, but rather from secondary mutations [or other forms of heritable phenotypic variation[31]] that accumulated at other loci. The total abundance of the expanding lineages was similar across mice in the same dietary conditions, even when the relative order of the lineages varied. However, we observed large variations across the different diets and even larger variations between the *Bacteroides* species. For example, the highlighted lineages in the HF/HS diet in Fig. 1c–f accounted for ~50% of the *Bt*-7330 and *Bo* populations on day 16 but comprised a much smaller fraction in *Bc* and *Bt*-VPI. This shows that the apparent rates of adaptation (as measured by the expansion of the largest Tn-Seq lineages) can vary between closely related commensal gut species, and even between strains of the same species.

The differences between species were much less pronounced on day 4, with no individual lineage reaching >5% frequency, and most remaining <0.01% (Fig. 1c–f). While low read counts make it difficult to follow these lineages individually, we reasoned that their collective behavior could still encode information about the evolutionary forces operating on these shorter timescales. As a first step, we focused on the subset of lineages that were present at a given frequency $f_0$ in the

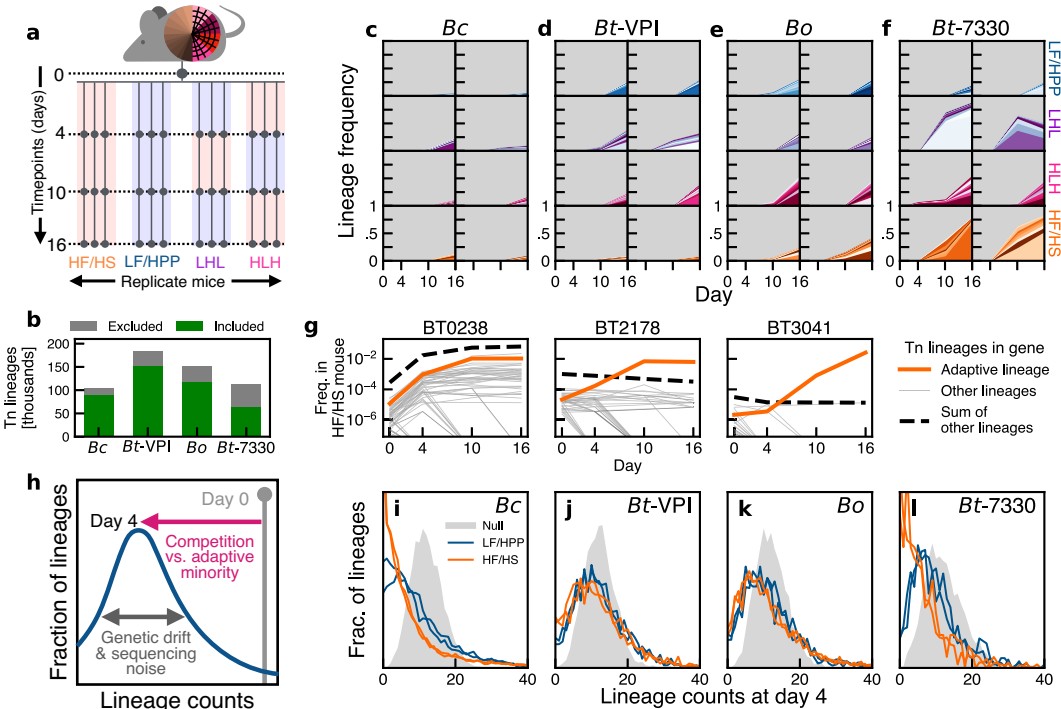

**Fig. 1 | Collective behavior of Tn lineages reveals rapid in vivo evolution in gnotobiotic mice. a** Schematic of Tn-Seq experiment in ref. 29. Mutant libraries of 4 *Bacteroides* strains and 11 other species were introduced in gnotobiotic mice fed different diets. **b** Number of Tn lineages included in analysis (see the "Methods" section). **c–f** Frequency trajectories of the 10 largest lineages at day 16 in two mice in each diet. **g** Frequency trajectories of all Tn lineages in three representative *Bt*-VPI genes in a single HF/HS mouse. Most lineages in *BT0238* expand, implying a beneficial effect of the gene knockout. In other genes (e.g. *BT2178* and *BT3041*), single lineages diverge in frequency. **h** Schematic of the evolutionary null model (see the "Methods" section), where neutral lineages decline due to both competition with fitter lineages and stochastic fluctuations from genetic drift. **i–l** Distribution of lineage read counts on day 4 for lineages with similar initial frequencies in the input library, measured in the four fixed-diet mice in (**b–e**). Gray distributions show the null expectation from sequencing noise alone (see the "Methods" section).

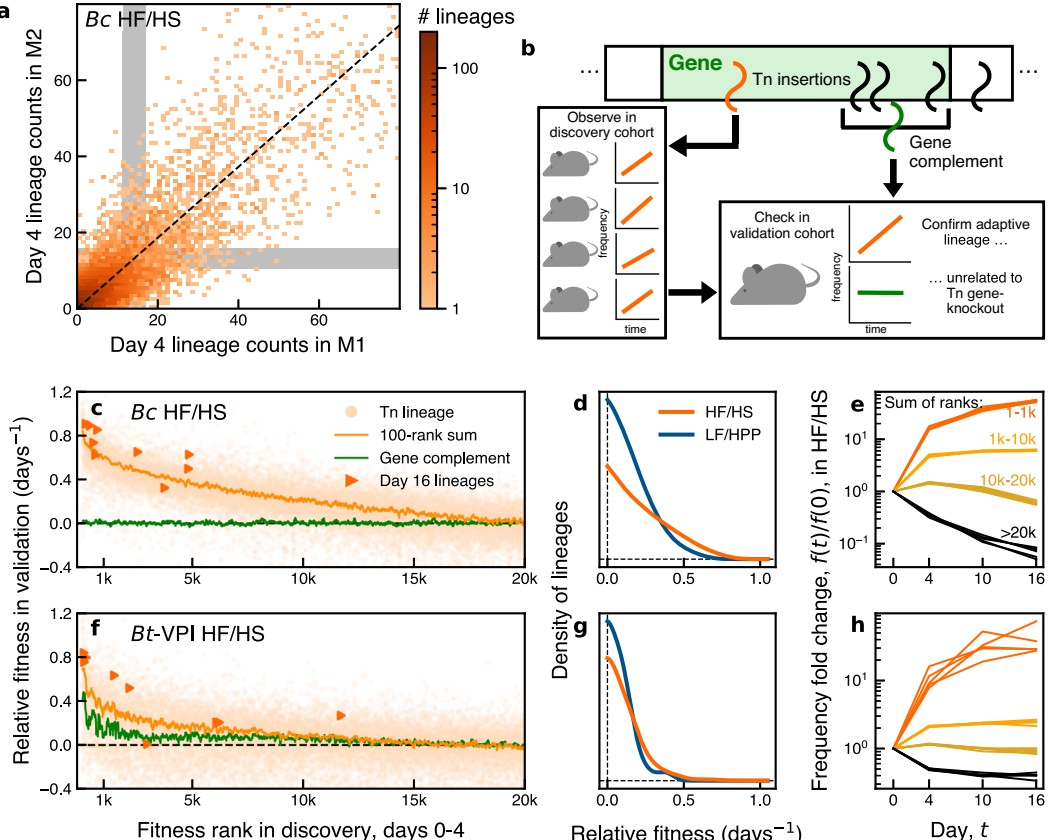

**Fig. 2 | Positive selection on thousands of lineages that are unrelated to their original gene knockouts. a** Joint distribution of day 4 read counts for a subset of *Bc* lineages in two representative mice. Lineages were drawn from a narrow range of initial frequencies in the input library (gray regions). **b** Schematic of cross-validation approach to detect adaptive lineages. **c, f** Average relative fitness in validation mice (*n* = 4) for the fittest 20,000 lineages in the discovery cohort (*n* = 5) in *Bc* (**c**) and *Bt-VPI* (**f**). Lines denote running averages of 100 lineages (orange) or their corresponding gene complements (green) (see the "Methods" section). Triangles indicate the 10 largest lineages in the HF/HS mice on day 16. **d, g** Fitness distribution of adaptive lineages inferred from panels **c** and **f** (see the "Methods" section) along with corresponding estimates for the LF/HPP diets (*n* = 4 discovery, *n* = 3 validation). **e, h** Groups of adaptive lineages continue to expand over time in 5 HF/HS mice. Colored lines are the total frequency of lineages in ranks indicated in (**e**).

initial library and examined the distribution of their frequencies at day 4 (Fig. 1h–l). In the simplest evolutionary null model (see the "Methods" section), the typical frequencies of these neutral lineages would decline due to competition with fitter variants in the population, as well as from stochastic fluctuations from genetic drift and sequencing noise (Fig. 1h; Supplementary Notes 1–3).

The observed distributions were largely consistent with this prediction. We found that the aggregated dynamics were remarkably similar across mice in the same diet, and to a lesser degree, between diets as well (Fig. 1i–l). However, we once again observed dramatic differences between the *Bacteroides* species. By day 4, most lineages remained close to their initial frequencies in *Bo* and *Bt-VPI*, while the majority of lineages substantially declined in *Bc* and *Bt-7330*. These differences could not have been caused by genetic bottlenecks, since we found that many of the same lineages were consistently present— and often expanded—in multiple independent mice (Fig. 2a). These correlations indicate that the collective behavior in Fig. 1i–l is not only driven by positive selection, but also that many of the causative variants must have been present in the initial Tn-Seq library prior to colonization of the mice. Such pre-existing variants have also been observed in other neutral barcoding systems[4,20–22,32], and are thought to arise from the multiple rounds of outgrowth and altered environmental conditions that are imposed during library creation[20,21].

Regardless of their source, we note that the apparent rates of adaptation in Fig. 1i–l are different from those in Fig. 1c–f. For example, *Bc* showed the strongest signatures of positive selection on day 4 but

had the smallest number of high-frequency lineages on day 16. *Bo* exhibited the opposite trend. This shows that the dynamics of common variants do not necessarily reflect the broader adaptation occurring within these populations at lower frequencies.

## Contrasting the spectrum of adaptive lineages in different gut species

To quantify this adaptive landscape more systematically, we sought to infer the fitnesses of the adaptive lineages driving the dynamics in Figs. 1 and 2a. The low read counts and fluctuating environmental conditions in this dataset make it difficult to apply existing methods[20,21], which assume that the fitness benefits of each lineage are constant and well-sampled across multiple consecutive timepoints. We therefore turned to a cross-validation approach that took advantage of the large number of biological replicates and the high levels of pre-existing variation implied by Fig. 2a.

To implement this approach, we first ranked each lineage by its average fold change across a subset of the mice on a given diet (the "discovery" cohort), and we compared this ranking to the average relative fitness in a held-out "validation" cohort from the same diet (Fig. 2b; see the "Methods" section). We reasoned that if the expanding lineages were driven by selection on preexisting variants, then their fitness in the validation cohort should be consistently positive as well. Figure 2c shows an example of this approach for the *Bc* populations in the HF/HS diet between days 0 and 4. While the fitnesses of the individual lineages were noisy as expected, we nevertheless observed a

clear enrichment in positive relative fitness among the top ~15,000 lineages (out of the ~90,000 we examined). By inverting the rank-order curve in sliding windows, we can obtain a self-consistent estimate for the distribution of fitnesses of the adaptive lineages that is robust to the presence of sampling noise (Fig. 2d, Supplementary Fig. 2; see the "Methods" section). These results suggest that >10% of the lineages in *Bc* experienced strong positive selection during the first four days of colonization (see the "Methods" section; Supplementary Note 4). This cohort continued to expand over the next 12 days (Fig. 2e), suggesting that the fitness benefits of these lineages were not confined to this initial time interval.

In principle, these in vivo fitness benefits could be caused by the gene-knockout effects of the original Tn insertions. Under this hypothesis, we would expect that the other lineages with insertions in the same gene should also expand over the same time interval (Fig. 2b). Surprisingly, however, we observed no strong correlation between the relative fitnesses of the putatively adaptive lineages in *Bc* and the fitnesses of their corresponding "gene complements" (Fig. 2c, see the "Methods" section). This suggests that their in vivo fitness benefits were caused by secondary mutations (or other forms of heritable genetic variation) that accumulated in the library prior to colonization. Our analysis shows that the fitness benefits of these mutations are large by evolutionary standards (>10% per day), and are comparable to the "fitness determinants" detected in the original transposon screen (Supplementary Fig. 3). We also observed considerable variation in fitness within the subset of adaptive lineages (Fig. 2c, e and Supplementary Fig. S4) suggesting that their benefits derived from different underlying mutations.

Similar signatures were present in the other diets and other *Bacteroides* species, though the number and magnitudes of the fitness benefits were somewhat different (Fig. 2f, g and Supplementary Figs. 5 and 6). For example, the number of strongly expanding lineages in *Bt-VPI* was lower than in *Bc*. Many of these lineages were also clustered in the same genes (e.g. *BT0238* in Fig. 1g), suggesting that their fitness benefits were caused by their original Tn insertions. However, even in this case, we found that loss-of-function variants accounted for only a small fraction of the putatively adaptive lineages since thousands of other lineages expanded by similar amounts (Supplementary Fig. 4). *Bo* and *Bt-7330* showed similar trends (Supplementary Fig. 5). In each of these cases, we found that the largest lineages at day 16 were enriched among the putatively adaptive mutations at day 4. Interestingly, however, these eventual winners were not necessarily the fittest lineages early on, suggesting that further mutations or environmental shifts were required to reach their dominant frequencies. This highlights how chance and competition among numerous low-frequency variants can play an important role in determining which mutations rise to appreciable frequencies within a host.

## Pleiotropic fitness tradeoffs across time and between diets

We next examined how this adaptive landscape varied over time and in different dietary conditions. The large number of pre-existing variants revealed by Fig. 2 provides a unique opportunity to address this question, by asking how the fitnesses of the same lineages co-vary across other diets and time intervals[28,33,34].

Despite broad differences in the shape of the adaptive spectrum across diets (Fig. 2d, g and Supplementary Fig. 6), we observed that the relative fitnesses of these lineages were remarkably consistent across the HF/HS and LF/HPP diets during the first four days of colonization (Fig. 3a, h–k and Supplementary Fig. 7). This indicates that the thousands of adaptive lineages we observed in Fig. 2c, f were not specific to the host diet. Intriguingly, the in vivo relative fitnesses in *Bc* were also highly correlated with in vitro fitnesses measured in several media (Supplementary Fig. 8), suggesting that they were also not specific to the complex features of their host environment.

In contrast, we found that the relative fitnesses of the lineages were only weakly correlated across time intervals in each of the *Bacteroides* species (Fig. 3b, h–k). This lack of correlation was not driven by the absence of selection at later times: Fig. 3h–k and Supplementary Fig. 9 show that the relative fitnesses during days 4–10 were still comparatively well correlated within the same diet. Instead, these results indicate that the selection pressures shifted over time, potentially driven by the rapid changes in the host environment or the surrounding microbial community. For example, the initial colonization of a germ-free gut could favor variants that are better able to survive the transit through the stomach or expand more rapidly in the cecum, while a fully saturated gut could select for variants that are beneficial during nutrient-limited competition.

Consistent with this hypothesis, we found that the relative fitnesses in later time intervals were only weakly correlated across diets (Fig. 3c, h–k), suggesting that host diet can have a substantial impact on selection at later times. Across the larger set of adaptive lineages, we identified hundreds of individual examples with strong fitness tradeoffs in different dietary conditions (Fig. 3d–f and Supplementary Fig. 10; Supplementary Note 5) Some lineages expanded 10-fold in LF/HPP, but were neutral or deleterious in HF/HS (Fig. 3d, e); other lineages displayed the opposite trend (Fig. 3f). These same lineages exhibited diverse behaviors in other time intervals as well. For example, while the example lineages in Fig. 3d, e displayed similar tradeoffs between days 4-10, only one of them expanded between days 0-4, while the other was effectively neutral. Conversely, the lineages in Fig. 3e, f were both effectively neutral between days 0–4 but exhibited opposing tradeoffs between days 4–10. This diverse range of behaviors provides further evidence that the adaptive lineages were driven by different underlying mutations, which can be differentially amplified by specific sequences of environments.

Despite these strong tradeoffs for individual lineages, our broader characterization revealed no strong evidence for a *global* tradeoff in the underlying fitness landscape. We found that many individual lineages consistently expanded in both diets (Fig. 3g and Supplementary Fig. 10), demonstrating that it is possible for evolution to improve fitness in both environments simultaneously. The lone exception was the comparison with the effective fitness during Tn library generation (see the "Methods" section), which was anti-correlated with in vivo fitness in *Bc* and *Bt-7330* (Supplementary Fig. 11). These limited anti-correlations suggest that the long-term tradeoffs observed at the population level[2] might not necessarily reflect an underlying physiological constraint, but may actually be an emergent property of their in vivo evolutionary dynamics[33].

A striking example of this behavior is illustrated by the handful of lineages that reached the largest frequencies by the end of the experiment. These lineages provide a proxy for the mutations that are likely to dominate the population at long times. We found that the largest lineages in the constant diets exhibited an apparent fitness tradeoff in *Bc*, with higher fitnesses in their home environment and average fitnesses in the other (Fig. 3c). In contrast, the alternating diets consistently selected for lineages that were fitter in both environments, despite their lower overall representation in the underlying fitness distribution (Fig. 3c and Supplementary Fig. 12). This illustrates how clonal competition and fluctuating selection pressures combine to determine the emergent fitness tradeoffs within a population.

## Discussion

Together, these results show how the fine-scale dynamics of genome-wide transposon libraries can enable quantitative inferences of in vivo evolutionary forces. We found that the early stages of colonization can be dominated by intense competition between thousands of adaptive variants − most of which would not be observed with traditional whole-genome sequencing approaches. While we have observed these dynamics in native human gut strains, it is possible that the high rates

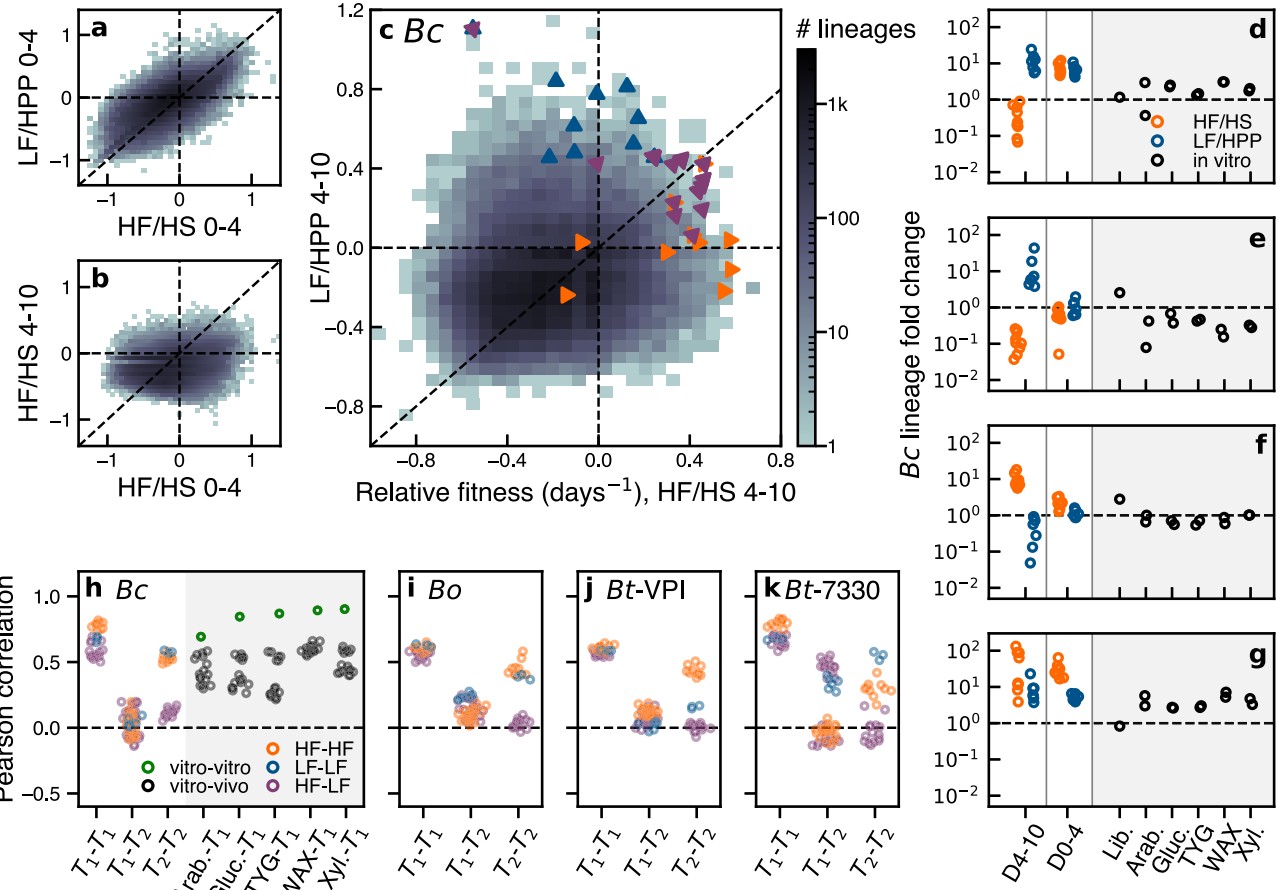

**Fig. 3 | Selection pressures shift over time to reveal diet-dependent fitness tradeoffs.** a–c Joint distribution of relative fitnesses in (**a**) HF/HS vs. LF/HPP diets over days 0–4, (**b**) days 0–4 vs. 4–10 in the HF/HS diet, and (**c**) HF/HS vs. LF/HPP diets over days 4–10; triangles indicate the 10 largest lineages at day 16 in the HF/HS (orange), LF/HPP (blue), or alternating (purple) diets. **d**–**g** Example lineages with strong fitness tradeoffs in different in vivo and in vitro conditions; circles indicate independent mice or in vitro cultures. Mice fed alternating diets are grouped with their respective diet at each time interval. **h**–**k** Pearson correlation coefficients of relative fitness values of lineages across different pairs of environments (see the "Methods" section). Symbols denote individual pairs of biological replicates. $T_1$ = days 0–4; $T_2$ = days 4–10; Lib. library creation, Arab. arabinose; Gluc. glucose; Xyl. xylose.

of adaptation we observed here could be driven by the novelty of the murine host or the comparatively low diversity of the artificial gut community. Our approach could be used to test these hypotheses in future experiments, by examining how the spectrum of fitness benefits differs in communities with higher levels of taxonomic diversity[35].

A key limitation of this approach is that it does not provide direct information about the genetic targets of adaptation. Future experiments could begin to map these molecular drivers by isolating and sequencing a subset of the adaptive lineages[36], though this would require a substantial sequencing effort, involving hundreds of isolates, to thoroughly sample the adaptive diversity in each population (Supplementary Fig. 13; Supplementary Note 6). Figure 3 also suggests that it may be possible to cluster the phenotypic impacts of these variants directly, by examining their pleiotropic tradeoffs across a large panel of environmental conditions[28,30] (Fig. 4a). For example, using the existing environments in Fig. 3, we can identify subsets of adaptive lineages whose pleiotropic tradeoffs strongly resemble the gene-level profiles of unrelated genes (Fig. 4b, c; see the "Methods" section). This suggests that the fitness benefits of these lineages were caused by loss-of-function mutations in the associated genes or pathways, providing a link to their underlying function.

Using this approach, we identified a cluster of 235 adaptive lineages in *Bt-VPI* that strongly resembled loss-of-function mutations in the polysaccharide utilization locus *PUL66* (Fig. 4b), which encompasses genes *BT3698-3705* that are responsible for starch

metabolism[37–39]. The disruption of *PUL66* was previously shown to be a fitness determinant in the HF/HS diet in the original Tn-Seq screen[29]; Fig. 4b shows that Δ*PUL66*-like phenotypes are also easily accessible via secondary mutations (though they are still not the fittest lineages in the population). A contrasting example is provided by the *BTO238* gene (Fig. 4c), which encodes an anaerobic sulfatase-maturating enzyme (anSME) that is important for the utilization of mucin and other sulfated hosts glycans[40–42]. While the disruption of *BTO238* provided significantly larger fitness benefits than Δ*PUL66* in the HF/HS diet (equivalent to an extra ~8-fold increase by day 16), we observed only a handful of adaptive lineages that exhibited fitness profiles similar to Δ*BTO238* (Fig. 4c). The larger number of Δ*PUL66*-like lineages allowed them to collectively account for a larger fraction of the *Bt-VPI* population at the end of the experiment (~3% vs. <1%), despite their smaller initial fitness benefits. This illustrates how differences in mutational accessibility can play a crucial role in determining the phenotypic response of an evolving population.

In addition to these annotatable examples, we also identified recurrent targets of selection whose fitness profiles were distinct from any gene-level knockouts (Fig. 4d, e and Supplementary Fig. 15; see the "Methods" section). These novel phenotypic clusters were particularly prevalent in *Bc*, which exhibited larger numbers of adaptive lineages (and fewer beneficial gene knockouts) compared to *Bt-VPI*. These different examples in Fig. 4 and Supplementary Fig. 15 show how the secondary mutations identified by our approach can illuminate

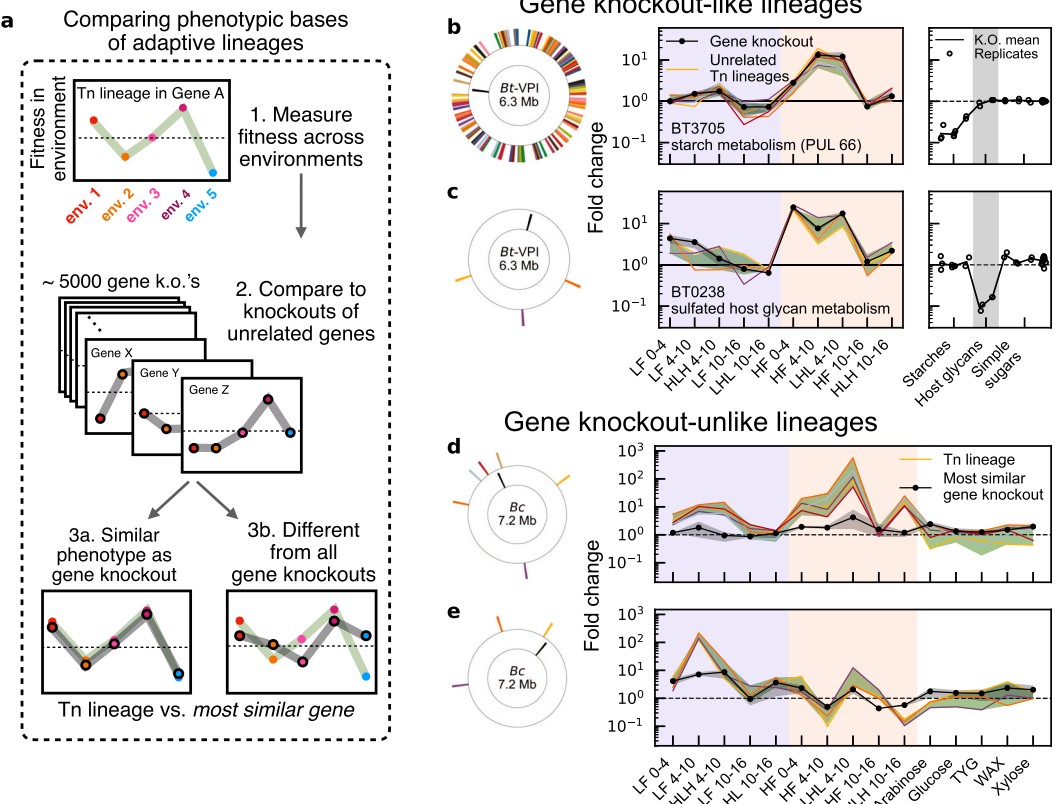

**Fig. 4 | Inferring the functional targets of adaptation using pleiotropic fitness tradeoffs across many environmental conditions. a** Fitness profiles of adaptive lineages across environments can be used to identify clusters of phenotypically similar lineages (see the "Methods" section). **b**, **c** Groups of adaptive lineages in *Bt-VPI* that resemble loss-of-function mutations in other genes. Left panels show the location of the target gene (inner ring, black) relative to unrelated Tn lineages (colors). Middle panels show the estimated fitness profiles (solid curves, see the "Methods" section) across in vivo environments (diet + time interval); for clarity, at most 5 representative lineages are shown for each cluster. Green bands denote the

interquartile (**b**) or full (**c**) ranges among all lineages in the cluster, and black bands are the range of knockout profiles from a random two-part split of data (see the "Methods" section). Right panels show fold-change of the same knockout in in vitro conditions measured in another study[30]. **d**, **e** Examples of novel phenotypic clusters in *Bc* that do not resemble any gene knockout. Fitness profiles are the same as in (**b**, **c**); green bands represent the full range. In both examples, the best-matching gene (black, see the "Methods" section) is >30 kb away from any lineage in the cluster.

regions of the adaptive landscape that are not accessible via traditional knockout screens.

Much of our analysis has been made possible by the presence of standing genetic variants, which appear to dominate the dynamics over the timescales we have considered. While pre-existing variants are also observed in other neutral barcoding systems[4,20–22,32], it remains unclear whether the higher rates we observed here are driven by specific features of the Tn-Seq protocol in ref. 29 (e.g. antibiotic selection or aerobic conjugation) or the increased importance of non-point-mutation processes (e.g. phase variation) that occur at high rates in some *Bacteroides* species[43–45]. In either case, the phenotypic diversity revealed by our analysis shows that these standing variants are not all caused by a single high-rate mutation (e.g. self-diploidization in ref. 36), but instead comprise a broader adaptive landscape that can drive in vivo evolution.

Finally, while we have focused on the dominant signals of selection on standing variants, it is possible to extend our approach to identify signatures of de novo mutations (Fig. 5a, b and Supplementary Fig. 16; see the "Methods" section) and rates of genetic drift (Fig. 5c–e and Supplementary Fig. 17; see the "Methods" section) by examining the deviations around this dominant trend. Taken together, these results suggest that future applications of genetic barcoding—combined with quantitative evolutionary modeling—could be a promising tool for resolving in vivo evolutionary forces in complex microbial communities.

## Methods
### Data and preprocessing
Raw data were obtained from a previous study[29], in which transposon libraries of 4 human *Bacteroides* strains (*Bc*, *Bo*, *Bt-VPI*, and *Bt-7330*) were combined with 11 other species and gavaged into gnotobiotic mice. Transposon libraries were created using the same protocol[29], which included 6 h of mating under aerobic conditions and 2 days of growth on antibiotic-treated plates (50 µg/mL erythromycin and 200 µg/mL gentamicin). Multi-taxon transposon insertion sequencing (InSeq) was performed on each input library (with 23–41 technical replicates per species), as well as on fecal samples taken on days 4, 10, and 16; we focused on the 16 replicate mice that had sufficient InSeq data at all 3 timepoints. One of the species (*Bc*) was also assayed in a variety of in vitro conditions. The full list of samples and technical replicates is provided in Supplementary Data 1.

Raw sequencing reads were mapped to reference genomes for each of the 4 *Bacteroides* strains. After removing the transposon sequence, each read was matched to its corresponding insertion location on the reference genome using a custom Python script (Supplementary Code 1). Only exact matches were retained, and reads that matched to multiple locations were excluded. We assumed that each unique location $\ell$ corresponded to a distinct lineage founded by a single transposon insertion event, and we calculated the total number of reads $R_{\ell,s}$ corresponding to lineage $\ell$ in sample $s$.

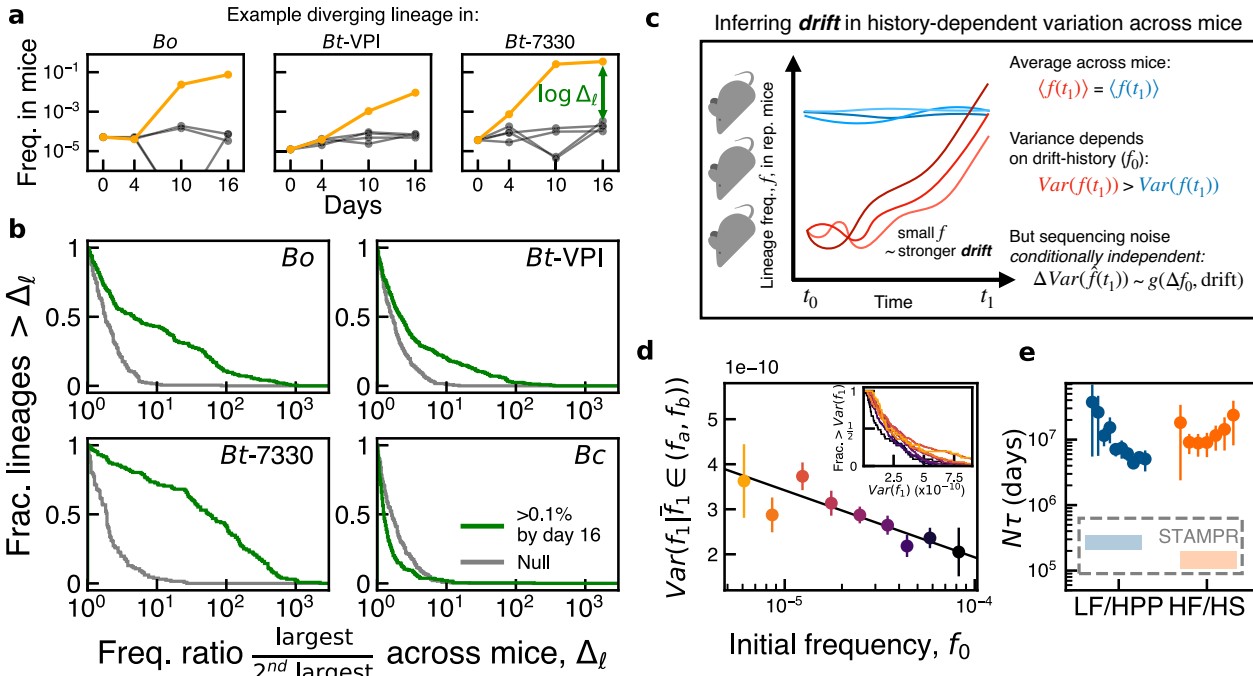

**Fig. 5 | Signatures of de novo mutations and genetic bottlenecks from the variability across replicates. a** Examples of Tn lineages that expanded in one HF/HS mouse (orange) but not in others (black). The ratio between the largest and second-largest frequencies defines a divergence score ($\Delta_\ell$) that can signal the acquisition of a beneficial mutation (see the "Methods" section). **b** Distribution of divergence scores on day 16 for all lineages that reached >0.1% frequency in at least one HF/HS mouse (green), versus a random set of noise-matched controls (gray; see the "Methods" section). The enrichment of large divergence scores in *Bo*, *Bt*-VPI, and *Bt*-7330 (but not *Bc*) suggests that de novo mutations play a greater role in some species than others. **c**–**e** Measuring genetic bottlenecks in the presence of widespread fitness variation. **c** Population genetic theory predicts that for a given current frequency, lineages that expanded more rapidly in the recent past will exhibit more variation across replicates since they had a smaller size at the initial timepoint ($f_0$); by contrast, sampling noise at the current timepoint is conditionally independent of $f_0$. **d** An example of this signature in *Bc*, showing the variance in day 4 frequency across LF/HPP mice for lineages with similar mean frequencies ($7 \times 10^{-5} \leq \bar{f}_1 \leq 8 \times 10^{-5}$) but different values of $f_0$. Points denote means and boot-strapped standard errors of different initial frequency bins (33–256 lineages per bin, 1063 total), while inset shows the full distributions. The significant correlation with $f_0$ is consistent with genetic drift (one-sided $p < 10^{-4}$, permutation test; see the "Methods" section). **e** Inverting the regression in (**d**) yields an estimate of the effective population size $N_e \tau_e$, where $\tau_e$ is the effective generation time (see the "Methods" section). Symbols denote least-squares estimates and standard errors (see the "Methods" section) for *Bc* populations obtained from different final frequency ranges (ordered left to right from smallest to largest, representing 669–6391 lineages). For comparison, shaded regions denote a range of estimates obtained from STAMPR[51].

To arrive at a conservative set of quasi-neutral Tn lineages for each *Bacteroides* strain, we removed from downstream analysis all lineages whose corresponding transposons fell within or 100 bp upstream of any of the "fitness determinant" genes previously identified by Wu et al.[29]. Most of these gene knockouts had deleterious fitness effects, though a small fraction were beneficial (see Tables S4A-B, S9A-B, S14A-D of ref. [29]). This filtering step left a total of $n = 418{,}879$ lineages across the four libraries (88,396 in *Bc*; 117,020 in *Bo*; 150,849 in *Bt-VPI*; and 62,614 in *Bt-7330*), which we used for all of our subsequent analysis. We estimated the relative frequencies of these remaining lineages using the plug-in estimator,

$$\hat{f}_{\ell,s} = \frac{R_{\ell,s}}{\sum_{\ell'} R_{\ell',s}}, \qquad (1)$$

where the denominator sums over all of the filtered lineages within a given *Bacteroides* strain. This renormalization scheme ensures that the relative fitnesses inferred in our later analyses are independent of the fitness determinant genes examined in ref. [29]. (The exceptions are Fig. 4 and Supplementary Figs. 3, 14, and 15, which compare the relative fitnesses of the fitness determinant genes identified by Wu et al.[29] with the additional adaptive lineages identified in the present work.)

### Model of evolutionary dynamics
We assumed that the temporal dynamics of the Tn lineages could be described by a simple evolutionary model, in which the lineages within

a given mouse $m$ competed with each other as a well-mixed population. The frequencies of rare lineages ($f_{\ell,m} \ll 1$), measured with respect to the subpopulation of filtered lineages, are then described by a system of coupled stochastic differential equations,

$$\frac{\partial f_{\ell,m}}{\partial t} = [s_{\ell,m}(t) - \overline{X}_m(t)]f_{\ell,m} + \sqrt{\Lambda_m(t)f_{\ell,m}} \cdot \eta_{\ell,m}(t), \qquad (2a)$$

where $\eta_{\ell,m}(t)$ is a Brownian noise term with mean zero and variance one[46], and $\Lambda_m(t)$ is the strength of genetic drift in mouse $m$ at time $t$[47]. Each lineage $\ell$ has instantaneous fitness $s_{\ell,m}(t)$, while $\overline{X}_m(t)$ is the mean fitness of the filtered lineages,

$$\overline{X}_m(t) = \sum_{\ell} s_{\ell,m}(t) \cdot f_{\ell,m}(t). \qquad (2b)$$

Equation (2) is a time-dependent generalization of the branching process model employed in previous in vitro lineage tracking studies[20,21]. The additional time dependence accounts for shifting selection pressures and population bottlenecks that might arise in more complex in vivo settings (e.g. due to changes in the host environment or in the composition of surrounding community). In Supplementary Note 1, we show that this model is robust to the exclusion of certain lineages (i.e. those falling in fitness determinant genes), provided that $\overline{X}_m(t)$ and $\Lambda_m(t)$ refer to the subset of retained lineages.

The deterministic component of Eq. (2) depends on the time-dependent fitness of the lineage, $s_{\ell,m}(t)$, and the mean fitness of the population, $\overline{X}_m(t)$. Previous lineage tracking studies have sought to decompose these contributions using a pre-defined or inferred set of neutral lineages[20,21,48]. Identifying such neutral lineages is difficult in our present dataset, due to the large fraction of non-neutral variation present in these libraries. Instead, we worked directly with relative fitness $x_{\ell,m}(t) \equiv s_{\ell,m}(t) - \overline{X}_m(t)$, which measures the instantaneous growth rate of a lineage as it competes with its surrounding population. We also defined a time-averaged version of the relative fitness over a given time interval $t_0 \le t \le t_1$:

$$\chi_{\ell,m,t_0:t_1} = \frac{1}{t_1 - t_0} \int_{t_0}^{t_1} [s_{\ell,m}(t) - \overline{X}_m(t)]\mathrm{d}t , \quad (3)$$

which was the focus of our downstream analysis.

## Model of sampling noise from sequencing

We assumed that the observed read counts $R_{\ell,s}$ were generated from the within-host dynamics in Eq. (2) through an additional sampling process, which encapsulates the combined effects of cell sampling, PCR amplification, and DNA sequencing. We assumed that this sampling process is unbiased, so that given a true frequency $f_{\ell,m_s}(t_s)$ in fecal sample $s$ of mouse $m_s$ at time $t_s$, the average number of observed read counts is given by

$$\langle R_{\ell,s} | f_{\ell,m_s}(t_s) \rangle = D_s \cdot f_{\ell,m_s}(t_s) , \quad (4)$$

where $D_s$ is the total coverage of sample $s$. Similarly, the variance can be written in the general form

$$\mathrm{Var}(R_{\ell,s} | f_{\ell,m_s}(t_s)) = \kappa_s \cdot D_s \cdot f_{\ell,m_s}(t_s) , \quad (5)$$

where $\kappa_s$ is a constant that describes the deviations from simple Poisson sampling. While the full distribution of the sampling process can be complicated, we follow previous work and assume that it can be approximated by a second branching process[20,21]. In Supplementary Note 2, we show that this sampling process combines with the within-host dynamics in Eq. (2) to produce an effective branching process with mean and variance:

$$\langle R_{\ell,s} \rangle = D_s \cdot a_{\ell,m_s}(t_s) ,$$
$$\mathrm{Var}(R_{\ell,s}) = D_s^2 \cdot a_{\ell,m_s}(t_s) \left[ \frac{\kappa_s}{D_s} + 2 b_{\ell,m_s}(t_s) \right] , \quad (6a)$$

where the functions $a_{\ell,m}(t)$ and $b_{\ell,m}(t)$ are defined by

$$a_{\ell,m}(t) = f_{\ell,m}(t_0) \cdot e^{\int_{t_0}^{t} x_{\ell,m}(t')\mathrm{d}t'} ,$$
$$b_{\ell,m}(t) = e^{\int_{t_0}^{t} x_{\ell,m}(t')\mathrm{d}t'} \int_{t_0}^{t} \mathrm{d}t' \frac{\Lambda_m(t')}{2} e^{-\int_{t_0}^{t'} x_{\ell,m}(t'')\mathrm{d}t''} . \quad (6b)$$

Equation (6) links the observed read counts $R_{\ell,s}$ to the underlying evolutionary parameters $\Lambda_m(t)$, $s_{\ell,m}(t)$, and $\overline{X}_m(t)$ in each mouse. All analyses were derived by considering different limits of this basic model.

## Distributions of lineage frequency shifts over time

The theoretical predictions for the lineage frequency shifts in Fig. 1h were derived from the branching process model in Eq. (6) (Supplementary Note 3). To compare these predictions with the data, we tabulated the empirical distributions of day 4 read counts for the subset of lineages in each mouse whose day 0 frequencies fell in the range $10/D_{m,4} \le f_0 \le 15/D_{m,4}$ (Fig. 1i–l). These day 0 frequencies were estimated by pooling all but one of the technical replicates of the input library. As a comparison, we performed the same procedure on the remaining input replicate to obtain an empirical null distribution showing the effects of technical noise alone (Fig. 1i–l). Deviations from this null distribution suggest that the observed dynamics are driven by the evolutionary forces of natural selection and/or genetic drift.

The joint distribution in Fig. 2a was computed using a similar procedure. We identified a subset of lineages with similar initial frequencies, $2 \times 10^{-5} < f_0 < 3 \times 10^{-5}$ in the input library, and measured their day 4 read counts in a pair of mice in the same diet. Under our simple model above, this joint distribution should factor into a product of the two marginal distributions $[p(R_1, R_2) \approx p(R_1)p(R_2)]$, regardless of their individual locations or widths. The strong correlations in Fig. 2a indicate departures from this simple model, in which a substantial fraction of the focal lineages have non-zero fitnesses that are shared across independent mice.

## Estimating relative fitnesses from read count trajectories

We estimated the relative fitnesses in Eq. (3) by pooling observations from multiple replicate mice in the same environment $e$. For a given cohort of mice $\mathcal{M}$, we estimated the time-averaged relative fitness using the plug-in estimator,

$$\hat{\chi}_{\ell,e,t_0:t_1} = \frac{1}{t_1 - t_0} \log \frac{\max\left\{ \bar{f}_{\ell,\mathcal{M}}(t_1), \min\{\frac{1}{D_{\mathcal{M}}(t_1)}, \bar{f}_{\ell,\mathcal{M}}(t_0)\} \right\}}{\max\left\{ \bar{f}_{\ell,\mathcal{M}}(t_0), \min\{\frac{1}{D_{\mathcal{M}}(t_0)}, \bar{f}_{\ell,\mathcal{M}}(t_1)\} \right\}} , \quad (7a)$$

where $\bar{f}_{\ell,\mathcal{M}}(t)$ is the weighted average of the lineage's frequency within the cohort,

$$\bar{f}_{\ell,\mathcal{M}}(t) \equiv \frac{\sum_{m \in \mathcal{M}} R_{\ell,(m,t)}}{\sum_{m \in \mathcal{M},\ell} R_{\ell,(m,t)}} \equiv \frac{R_{\ell,\mathcal{M}_i}}{D_{\mathcal{M}_i}} , \quad (7b)$$

and $D_{\mathcal{M}}(t)$ is the total coverage of the library across the cohort of mice,

$$D_{\mathcal{M}}(t) \equiv \sum_{\ell,m \in \mathcal{M}} R_{\ell,(m,t)} . \quad (7c)$$

When $\bar{f}_{\ell,\mathcal{M}}(t_0)$ and $\bar{f}_{\ell,\mathcal{M}}(t_1)$ are both greater than zero, Eq. (7) reduces to the standard log-ratio estimator, $\hat{\chi} \approx \frac{1}{t_1 - t_0} \log \bar{f}_{\ell,\mathcal{M}}(t_1)/\bar{f}_{\ell,\mathcal{M}}(t_0)$. On the other hand, if either of the frequencies are zero, the $\min\{\cdot\}$ terms act like an effective pseudocount, which is conservatively biased to assign zero relative fitness to lineages with sufficiently low frequency [e.g. $f(t_0) < 1/D(t_1)$].

We used a similar approach to estimate relative fitnesses in the in vitro environments (Fig. 3h–k and Supplementary Fig. 8). Wu et al.[29] completed the *Bc* library in 5 in vitro growth media; 2 independent cultures were inoculated in each medium, and 3 aliquots were sequenced from each culture after reaching stationary phase. We estimated the relative fitness in each independent culture using Eq. (7); since no explicit time interval was reported, we set $t_1 - t_0 = 1$ to reflect the duration of a typical overnight culture.

We also defined a special in vitro environment representing the library creation process. We assumed that all of the Tn lineages were initially founded by unique insertion events in a single cell so that their frequencies in the input library reflect the differential growth that occurred during the library creation process. We estimated the relative fitness in this effective environment using Eq. (7) with a uniform initial frequency,

$$f_{\ell,\mathcal{M}}(t_0) \equiv \frac{1}{\sum_{\ell} 1} , \quad (8)$$

and an arbitrary time interval $t_1 - t_0 = 1$.

## Cross-validation procedure

We adopted a cross-validation approach to distinguish genuine selection on adaptive lineages from biological or technical sources of noise. We split $k$ mice that were maintained in the same dietary environment $e$ into equally sized discovery ($\mathcal{M}_{D_e}$) and validation ($\mathcal{M}_{V_e}$) cohorts. The replicate measurements of the input library were also evenly divided between the two cohorts. We used this partitioning to generate independent estimates of the relative fitness of each lineage in the discovery and validation cohorts. We ranked each lineage by its fitness in the discovery cohort, and examined how the fitnesses in the validation cohort varied as a function of their rank $\rho(\ell)$ in the discovery cohort (Fig. 2c, f and Supplementary Fig. 5). Since the two cohorts have independent sources of technical and biological noise, systematic correlations between these two quantities can be used to distinguish genuine fitness differences from statistical fluctuations in read counts.

To increase the signal-to-noise ratio, we restricted our attention to lineages with initial frequencies $>10^{-6.5}$ (equivalent to ~10 reads in the pooled input library in each species). In addition, we only examined lineages with a minimum number of expected reads in the validation cohort:

$$\min\{D_{\mathcal{M}_{V_e},0}, D_{\mathcal{M}_{V_e},1}\} \cdot \max\{\bar{f}_{\ell,\mathcal{M}_{D_e,0}}, \bar{f}_{\ell,\mathcal{M}_{D_e,1}}\} > 5. \quad (9)$$

These filters were used to generate the rank-ordered fitness distributions in Fig. 2c, f and Supplementary Figs. 5–7.

To distinguish systematic trends from the noisy estimates of individual lineages (Fig. 2), we coarse-grained groups of lineages based on their relative fitness rank, $\rho(\ell)$, in the discovery cohort. For a given range of ranks $\rho_0 \leq \rho \leq \rho_1$, we defined the coarse-grained frequency $\bar{f}_{\rho_0:\rho_1,\mathcal{M}}(t)$ by summing over the individual lineage frequencies in Eq. (7b):

$$\bar{f}_{\rho_0:\rho_1,\mathcal{M}}(t) \equiv \sum_{\rho_0 \leq \rho(\ell) \leq \rho_1} \bar{f}_{\ell,\mathcal{M}}(t). \quad (10)$$

Under the simple model in Eq. (2), this coarse-grained frequency will grow as

$$\langle \overline{f}_{\rho_0:\rho_1,\mathcal{M}}(t_1) \rangle = \bar{f}_{\rho_0:\rho_1,\mathcal{M}}(t_0) \cdot \exp\left[\bar{\chi}_{\rho_0:\rho_1,e,t_0:t_1} \cdot (t_1 - t_0)\right], \quad (11)$$

where $\bar{\chi}_{\rho_0:\rho_1,e,t_0:t_1}$ is the average relative fitness,

$$\bar{\chi}_{\rho_0:\rho_1,e,t_0:t_1} = \frac{1}{t_1 - t_0} \log\left(\frac{\sum_{\rho_0 \leq \rho(\ell) \leq \rho_1} \bar{f}_{\ell,\mathcal{M}}(t_0) \cdot \exp\left[\chi_{\ell,e,t_0:t_1} \cdot (t_1 - t_0)\right]}{\sum_{\rho_0 \leq \rho(\ell) \leq \rho_1} \bar{f}_{\ell,\mathcal{M}}(t_0)}\right). \quad (12)$$

Positive values of $\bar{\chi}_{\rho_0:\rho_1,e,t_0:t_1}$ indicate that at least some lineages in the coarse-grained grouping have positive relative fitness. We estimated this coarse-grained relative fitness in the validation cohort using an analogous version of Eq. (7), in which $\bar{f}_{\ell,\mathcal{M}}(t)$ is replaced by $\bar{f}_{\rho_0:\rho_1,\mathcal{M}}(t)$. The orange (HF/HS) and blue (LF/HPP) lines in Fig. 2c, f and Supplementary Figs. 5–7 show the coarse-gained relative fitnesses in the validation cohort in sliding windows of 100 consecutive ranks. The consistently positive values observed at lower values of $\rho(\ell)$ indicate that many of the underlying lineages had positive relative fitness.

## Distinguishing gene-level and lineage-level fitness effects

We used a similar coarse-graining procedure to estimate the relative fitness of the gene complement of each lineage (Fig. 2b). The gene complement was defined for a focal lineage $\ell$ if its transposon insertion fell in the coding region or <100 bp upstream of an annotated gene. In this case, the gene complement $\mathcal{G}(\ell)$ was defined to be the collection of all other lineages (excluding the focal lineage) that also fell within the

same gene as $\ell$. This collection of lineages defines a coarse-grained frequency,

$$\bar{f}_{\mathcal{G}(\ell),\mathcal{M}}(t) = \sum_{\ell' \in \mathcal{G}(\ell)} \bar{f}_{\ell',\mathcal{M}}(t), \quad (13)$$

which represents the total frequency of all of the other lineages associated with the same gene. If the relative fitness of lineage $\ell$ was caused by the gene knockout effect of the original Tn insertion, then the dynamics of the gene complement $\bar{f}_{\mathcal{G}(\ell),\mathcal{M}}(t)$ should be statistically similar to the dynamics of the focal lineage $\bar{f}_{\ell,\mathcal{M}}(t)$. We tested this hypothesis by estimating the relative fitness of the gene complement of each lineage using an analogous version of Eq. (7), with $\bar{f}_{\mathcal{G}(\ell),\mathcal{M}}(t)$ replacing $\bar{f}_{\ell,\mathcal{M}}(t)$. The green lines in Fig. 2c, f and Supplementary Fig. 5 show sliding averages of the gene complement fitnesses in the validation cohort (calculated using the same coarse-graining scheme described above) as a function of the relative fitness rank of the focal lineage in the discovery cohort.

## Validation using simulations

To confirm that our filtering and cross-validation procedure could consistently estimate lineage fitnesses in a population, we simulated lineage dynamics in "biological replicates" under conditions mimicking our experiments. First, we generated an empirical distribution of lineage fitnesses during days 0–4 by averaging across the 9 mice fed the HF/HS diet in this interval in the dataset using Equation (7). These inferred fitnesses, along with the input frequencies, were used to initialize 9 identical populations. We then simulated these populations for 40 generations (~4 days) under a Wright–Fisher model with a fixed population size $N_e = 10^8$. Finally, to generate simulated sequencing libraries, we modeled the sampling and sequencing of each population as a sequence of two Poisson sampling steps, each equal to the empirical sequencing depth at day 4 (which varied across *Bacteroides* species and mice). We then performed the filtering and cross-validation procedures described above to produce rank-order curves from these simulated populations (Supplementary Fig. 2). We also performed the same procedure on simulated populations with zero fitness differences between lineages. These simulations confirmed that our cross-validation approach could reliably distinguish the presence or absence of fitness variation in a population.

## Measuring the distribution of relative fitnesses

Our simulations showed that the coarse-grained rank order curves reliably estimated the ground truth rank order curves in each of the scenarios we considered. This implies that the distribution of estimated relative fitnesses among coarse-grained lineages approximates the true distribution among individual lineages.

For visual clarity, we report smoothed distributions after applying Gaussian kernel density estimation to the set of coarse-grained relative fitnesses using the `stats.gaussian_kde()` function in the SciPy library[49], with a bandwidth equal to 0.3 times the standard deviation of the coarse-grained lineages. The results are shown for both simulations (Supplementary Fig. 2) and the observed data (Supplementary Fig. 6). In Supplementary Note 4, we present an alternative approach to count the total number of adaptive lineages, without coarse-graining.

## Inferring fitness tradeoffs across environments

To estimate the joint distributions of relative fitnesses across a pair of environments and/or timepoints (Fig. 3a–c and Supplementary Fig. 9), we began by splitting the samples into non-overlapping cohorts for each of the two environments, $e_1$ and $e_2$. This division ensured that statistical fluctuations in one environment did not influence relative fitness estimates in the other environment. We used Eq. (7) to estimate the relative fitness in each environment for all lineages that satisfied Eq. (9) in at least one of the two environments.

The pairwise correlations in Fig. 3h–k were estimated from a single biological replicate in each environment, and Pearson correlations were computed for different pairs of biological replicates. However, to avoid biases from our plug-in estimator, in each pair we calculated the correlation only among lineages measured at non-zero frequency in both time points in each replicate.

We defined a lineage as exhibiting a fitness tradeoff if its relative fitness $\chi_{\ell,e,t_0:t_1}$ had opposite signs in a pair of environments or time intervals. Several example lineages from *Bc* with large tradeoffs, consistently observed across replicate mice, are illustrated in Fig. 3d–g. In Supplementary Note 5, we generalize the cross-validation approach above to infer the presence of hundreds of lineages with fitness tradeoffs across diets, as well as "generalist" lineages with diet-independent fitnesses (Supplementary Fig. 10).

### Identifying clusters of phenotypically similar lineages

We used the fitness measurements across multiple environmental conditions to identify clusters of phenotypically similar lineages (Fig. 4). The basic algorithm is outlined below; further discussion of the intuition behind this approach can be found in Supplementary Note 6.

**Identifying adaptive lineages with knockout-like mutations.** We first searched for adaptive lineages whose fitness profiles (i.e., the joint fitness across multiple environments) resembled one of the loss-of-function variants from the original Tn-Seq screen[29]. We identified these "knockout-like" lineages by comparing the relative fitness profiles of individual Tn lineages (measured across the panel of in vivo environments defined below) against the gene-level profiles of unrelated genes. Beneficial lineages whose fitness profiles closely matched one of the gene-level profiles and had no gene-level signal of their own were classified as knockout-like lineages, whose secondary mutations might have occurred within the associated gene (or within its associated operon or molecular pathway).

To define our set of environments, we took advantage of the fact that the in vivo selection pressures varied over time and across host diets (Fig. 3). We therefore split the data into $N_{env} = 10$ in vivo environments, each of which was a combination of a time interval (Days 0–4, 4–10, or 10–16) and a particular host–diet history (LF/HPP, HF/HS, LHL, HLH). Since LF/HPP and LHL shared the same host–diet history up to day 4 (and likewise HF/HS and HLH), we treated these cases as the same environment between days 0–4. (In the case of *Bc* the list of environments was augmented with the in vitro environments.)

We defined the coarse-grained fitness profile of a set of lineages $\mathcal{L}$ as the vector of log-fold-changes, in each environment,

$$\mathbf{W}_{\mathcal{L}} = \left( W_{e_1}, W_{e_2}, \ldots, W_{e_{N_{env}}} \right) \equiv \left( \log \frac{\overline{f}_{\mathcal{L},e_{1,1}}}{\overline{f}_{\mathcal{L},e_{1,0}}}, \log \frac{\overline{f}_{\mathcal{L},e_{2,1}}}{\overline{f}_{\mathcal{L},e_{2,0}}}, \ldots, \log \frac{\overline{f}_{\mathcal{L},e_{N_{env},1}}}{\overline{f}_{\mathcal{L},e_{N_{env},0}}} \right),$$
$$(14)$$

where $\overline{f}_{\mathcal{L},e_{i,j}}$ represents the *j*th timepoint of the *i*th environment in a particular cohort of replicates $\mathcal{M}_{e_i}$, such that

$$\overline{f}_{\mathcal{L},e_{i,j}} = \sum_{\ell \in \mathcal{L}} \overline{f}_{\ell,\mathcal{M}_{e_i}},$$
$$(15)$$

where $\overline{f}_{\ell,\mathcal{M}_{e_i}}$ is defined in Eq. (7b). Except where noted, fitness profiles were measured using every replicate available in each environment. Because we are interested in the fitness profiles of beneficial gene knockouts, we measured these frequencies with respect to the whole library, including the fitness determinant genes identified by Wu et al.[29]; this differs from our analysis in the rest of the study, which excluded lineages that fell in these genes. We measured the similarity between two fitness profiles using the

Euclidean distance,

$$d(\mathbf{W}_{\mathcal{L}}, \mathbf{W}_{\mathcal{L}'}) \equiv \sqrt{\sum_{e_k} (W_{\mathcal{L},e_k} - W_{\mathcal{L}',e_k})^2}.$$
$$(16)$$

Other metrics, e.g. correlation coefficients, did not qualitatively impact the results.

Next, we curated a reference list of strongly beneficial gene knockouts. We measured the coarse-grained fitness profile $\mathbf{W}_{\mathcal{G}(\emptyset)}$ of every gene $\mathcal{G}$ using the Tn lineages falling in $\mathcal{G}$. The notation $\mathcal{G}(\emptyset)$ indicates that, unlike the gene complement in Eq. (13), no Tn lineages were excluded from this coarse-grained fitness profile. We restricted our attention to gene knockouts that exhibited a fold change >10 in at least one environment (i.e., $\max_{e_k} W_{\mathcal{G}(\emptyset),e_k} > \ln 10$), or a fold change >5 in at least two environments. Many of these could be driven by the expansion of a Tn lineage that had acquired an adaptive secondary mutation. Hence, for each of the remaining gene knockouts, we recalculated a set of "leave-one-out" fitness profiles across all $\mathcal{G}(\ell)$, each excluding a single lineage $\ell$ in the gene:

$$S_{\mathbf{W}_{\mathcal{G}}} = \{\mathbf{W}_{\mathcal{G}(\ell)}; \ell \in \mathcal{G}(\emptyset)\}.$$
$$(17)$$

If a single adaptive lineage drove the coarse-grained fitness of a gene, its exclusion should dramatically reduce the corresponding leave-one-out fitness profile of the gene. Conversely, if the gene's coarse-grained fitness profile was driven by many similar lineages, then leaving out any one lineage should not dramatically change the knockout fitness profile. Based on this logic, we removed from consideration any gene that contained a leave-one-out fitness profile that did not exhibit a fold-change > 5 in one environment or a fold-change >3 in two environments. This procedure eliminated a majority of the remaining genes; visual inspection of individual lineage trajectories confirmed that nearly all were driven entirely by the behavior of single Tn lineages (e.g. Fig. 1g).

We next identified lineages with similar fitness profiles to the remaining high-quality beneficial gene knockouts. We deemed a lineage $\ell_c$ to resemble the knockout of a particular gene if $d(\mathbf{W}_{\ell_c}, \mathbf{W}_{\mathcal{G}(\emptyset)}) < d^*$. We chose a distance threshold $d^* = 2$, which gave visually well-defined clusters and was consistent with the typical distance between well-measured lineages and their own coarse-grained gene knockouts ($d \simeq 2$–3). For each of the strongly beneficial gene knockouts, we computed Eq. (16) for candidate lineages that were measured in every environment and were located >100 kb from the target gene.

After assembling a preliminary cluster of knockout-like Tn lineages for each target gene, we checked whether these clusters were likely to have been driven by secondary mutations, as opposed to the direct effects of Tn insertions in other functionally related genes. To be conservative, if an off-target gene was represented by multiple lineages in a given cluster, we removed those lineages from the cluster, since there is a chance that they could be driven by the direct effect of a Tn insertion in a gene that is functionally related to the target gene. This filter removed some lineages from larger clusters like *PUL*66 (Fig. 4a), though the number of multiply-represented genes in these clusters was consistent with a null expectation in which knockout-like lineages were randomly distributed across the genome.

The results of this pipeline are shown for two example *Bt-VPI* genes in Fig. 4a, b; several others are shown in Supplementary Fig. 14. Additional checks against noise-driven clustering and plotting details are discussed after the next section.

**Identifying novel clusters of secondary mutations.** Loss-of-function mutations will likely comprise only a fraction of the adaptive landscape[2]. The phenotypic effects—and therefore fitness profiles—of other variants may be qualitatively distinct from any gene knockout. To identify recurrent selection on such novel and strongly beneficial secondary mutations, we reversed the approach used in the previous

section: we clustered individual lineages based on their own fitness profiles and then looked for clusters of similar lineages that were different from every gene-level profile.

To identify novel lineage clusters, we first curated a large list of well-measured and highly fit lineages in each *Bacteroides* library. We only considered lineages that had a measurable fitness in all environments (i.e. were present in at least one of the two-time points comprising the environment), and had large positive fitnesses in at least one environment, $\max_{e_k} W_{\ell,e_k} > \ln 10$. We applied bottom-up hierarchical clustering on the resulting lineages using the `AgglomerativeClustering()` function from the Scikit-learn library[50]. We used an average linkage criterion with the distance metric Eq. (16) and a maximum merge distance of $\epsilon = 4$. For each resulting high-fitness cluster, $\mathcal{C}_\nu$, we defined the cluster-averaged profile as

$$\mathbf{W}_{\mathcal{C}_\nu} = \frac{\sum_{\ell \in \mathcal{C}_\nu} \mathbf{W}_\ell}{\sum_{\ell \in \mathcal{C}_\nu} 1} . \tag{18}$$

Since the initial high-fitness filtering omitted many adaptive lineages, we remapped all remaining lineages with fitness measurements in all environments to their nearest high-fitness cluster. For each lineage $\ell$, we found its distance-minimizing cluster $\mathcal{C}_\mu$:

$$\mathcal{C}_\mu = \underset{\mathcal{C}_\nu}{\mathrm{argmin}}\ d(\mathbf{W}_\ell, \mathbf{W}_{\mathcal{C}_\nu}) . \tag{19}$$

If this distance-minimizing cluster had $d < \epsilon$, the lineage was added to the cluster. After adding all such lineages, average profiles $\mathbf{W}_{\mathcal{C}_\nu}$ were recalculated over the augmented clusters.

We applied several stringent filters to discard clusters plausibly composed of lineages that acquired "knockout-like" secondary mutations discussed above. We excluded from further consideration clusters of size $k > 1$ if any gene was represented by more than $\max\{1, \lfloor \log_{10}(k) \rfloor\}$ lineages in the cluster since this should be rare for randomly distributed lineages (<1%). We also excluded clusters with only one lineage, since these could in principle be driven by a site-specific effect of its Tn insertion.

Finally, we directly compared the remaining clusters to gene knockout profiles. To prevent single lineages with strongly beneficial secondary mutations from biasing the measured gene profiles, we first compared the leave-one-out profiles in Eq. (17) for each gene: if k-means clustering with $k = 2$ isolated a single outlying profile, the corresponding lineage was removed and the gene profile was re-estimated. We then randomly partitioned the remaining Tn lineages in each gene into two equal-sized sets, and measured gene fitness profiles $\mathbf{W}_{\mathcal{G}}^{(1)}$ and $\mathbf{W}_{\mathcal{G}}^{(2)}$ from the coarse-grained frequencies of the two subsets. We did not further consider the minority of genes with unmeasured fitnesses in either subset. We computed, for each cluster $\mathcal{C}_\nu$ and gene $\mathcal{G}$, the distance $D_{\mathcal{C}_\nu,\mathcal{G}} \equiv d(\mathbf{W}_{\mathcal{C}_\nu}, \frac{1}{2}(\mathbf{W}_{\mathcal{G}}^{(1)} + \mathbf{W}_{\mathcal{G}}^{(2)}))$ (excluding from $\mathbf{W}_{\mathcal{G}}^{(i)}$ any lineage that was shared between $\mathcal{C}_\nu$ and $\mathcal{G}$). The use of the subset-averaged gene profile $\frac{1}{2}(\mathbf{W}_{\mathcal{G}}^{(1)} + \mathbf{W}_{\mathcal{G}}^{(2)})$ reduced the influence of any adaptive secondary mutations that had not been filtered out in the previous step (these replicates were also used to check against ascertainment bias, below). Clusters with large values of $D_{\mathcal{C}_\nu,\mathcal{G}}$ across all genes $\mathcal{G}$ were candidates for "novel" phenotypic clusters. Examples in several species (Fig. 4c, d and Supplementary Fig. 15) show the identified clusters and their most similar looking genes (i.e., those with smallest $D_{\mathcal{C}_\nu,\mathcal{G}}$).

**Controlling for ascertainment bias during clustering.** For both novel and gene knockout-like clusters, we checked that our results did not arise from ascertainment biases arising from the clustering of thousands of noisy lineage profiles. Biological replicates were split in each environment into $\mathcal{M}_1$ and $\mathcal{M}_2$, representing two independent sets of replicates across environments. For each lineage, we computed fitness profiles $\mathbf{W}_\ell^{(\mathcal{M}_1)}$ and $\mathbf{W}_\ell^{(\mathcal{M}_2)}$, constituting independent measurements of

the fitness profile that should be similar to one another, as well as to other $\mathbf{W}_\ell^{(\mathcal{M}_i)}$ in the cluster. We calculated the unweighted average of the two replicate profiles, $\langle \mathbf{W}_\ell \rangle = \frac{1}{2}(\mathbf{W}_\ell^{(\mathcal{M}_1)} + \mathbf{W}_\ell^{(\mathcal{M}_2)})$, for each lineage. The interquartile range of $\langle \mathbf{W}_\ell \rangle$ among co-clustered lineages is represented by the green shaded bands in Fig. 4 and Supplementary Figs. 14 and 15; if a cluster had fewer than 10 lineages, the full range is plotted instead. Similarly, the black-shaded regions in Fig. 4 and Supplementary Figs. 14 and 15 represent the spread between $\mathbf{W}_{\mathcal{G}}^{(1)}$ and $\mathbf{W}_{\mathcal{G}}^{(2)}$, described in the previous subsection.

## Signatures of additional de novo mutations

To identify signatures of de novo mutations, we focused on the fact that de novo mutations would initially arise within a single Tn lineage within a single mouse. If any of these mutations survives genetic drift, it should sweep through its Tn lineage and continue to spread through the population, leading to a lineage trajectory that diverged from the other biological replicates, as in Fig. 5a.

To examine the evidence for such mutations, we ranked the day 16 frequencies of each lineage $\ell$ across all $k = 5$ mice in the HF/HS diet, $f_{\ell,1}(16) \geq f_{\ell,2}(16) \geq \ldots \geq f_{\ell,k}(16)$. We defined a divergence metric $\Delta_\ell$ as the ratio between the largest and second-largest frequencies,

$$\Delta_\ell \equiv \frac{f_{\ell,1}(16)}{f_{\ell,2}(16)} . \tag{20}$$

Large values of $\Delta_\ell$ would be consistent with a beneficial mutation occurring in mouse 1, but may also be driven by biological or technical noise. To mitigate noise, we restricted our attention to lineages that reached > 0.1% frequency in at least one mouse (which required that they were sampled in at least 10 reads). We compared the observed distribution of $\Delta_\ell$ values to a null distribution obtained from a random set of lineages constructed to have a similar distribution of initial (day 0) frequencies, and measured at day 16 in at least two mice (Fig. 5b and Supplementary Fig. 16).

## Estimating the strength of genetic drift

To quantify the strength of genetic drift in vivo, we again focused on the variation across replicates, defined by

$$\hat{\mathrm{Var}}(\hat{f}_\ell(t))_{\mathcal{M}_2} = \frac{1}{|\mathcal{M}_2| - 1} \sum_{m \in \mathcal{M}_2} \left( \hat{f}_{\ell,m}(t) - \frac{1}{|\mathcal{M}_2|} \sum_{m' \in \mathcal{M}_2} \hat{f}_{\ell,m'}(t) \right)^2 . \tag{21}$$

We considered a simple version of Eq. (2), in which the relative fitness and strength of genetic drift are approximately constant over the relevant time interval. This implies that $\chi_{\ell,e,t} \approx \chi_{\ell,e}$ and $\Lambda_m(t) \simeq 1/N_e \tau_e$ for each mouse $m$ in environment $e$, where $N_e$ is the effective population size and $\tau_e$ is the effective generation time[47]. Given these assumptions, the expectation of Eq. (21) can be written in the form

$$\langle \hat{\mathrm{Var}}(\hat{f}_\ell(t))_{\mathcal{M}_2} \rangle = \underbrace{\frac{1}{|\mathcal{M}_2|} \sum_{m \in \mathcal{M}_2} \frac{\kappa_{m,t}}{D_{m,t}} \cdot \langle f_\ell(t) \rangle}_{\text{technical noise}} + \underbrace{\frac{t}{N_e \tau_e} \cdot \frac{\langle f_\ell(t) \rangle \left[ \langle f_\ell(t) \rangle / f_\ell(0) - 1 \right]}{\log \left[ \langle f_\ell(t) \rangle / f_\ell(0) \right]}}_{\text{genetic drift}} . \tag{22}$$

which we derive in Supplementary Note 7. This shows that the contributions to the variance from genetic drift and technical noise can be distinguished by their different scaling as a function of $\langle f_\ell(t) \rangle$ and $f_\ell(0)$. In particular, conditioned on the same expected final frequency $\langle f_\ell(t) \rangle$, lineages that began at smaller initial frequencies (implying a higher fitness) should have a higher variation in final frequency across replicates than those that began at larger initial frequencies.

To measure this correlation, we split replicate mice from the same diet into two equal-sized cohorts, $\mathcal{M}_1$ and $\mathcal{M}_2$. We estimated the mean

frequency of each lineage at time $t$ using the mice in $\mathcal{M}_1$, and we estimated the variance at time $t$ using the mice in $\mathcal{M}_2$. Estimating these quantities using separate cohorts prevents spurious, sampling-induced correlations between the estimates of the first and second moments. Based on these estimates, we selected lineages whose mean $\bar{f}_{\ell,\mathcal{M}_1}(t)$ fell within a narrow final frequency range $(f_{\min}(t), f_{\max}(t))$, and a broad initial frequency range $(f_{\min}(0), f_{\max}(0))$.

We repeated this process across all possible partitions of the mice (of equal size), concatenating lineages and their partition-specific estimated frequencies and variances. Combining all possible partitions effectively averages over the random partitions of replicate mice, while self-consistently expanding the number of lineages included in downstream analysis. However, most lineages pass filtering in multiple partitions, and frequency and variance estimates are correlated across splits. To remove these latent correlations, we randomly downsampled each lineage's mean and variance estimates to those of a single partition $\mathcal{M}_1^{(\ell)}$, $\mathcal{M}_2^{(\ell)}$, which was included in the initial concatenation. This concatenation-and-downsampling scheme generates a set of vectors $\mathcal{L} = \{W, X, Y\}$ representing the initial frequencies and estimated final frequencies and variances of included lineages, respectively:

$$
\begin{aligned}
W_\ell &= f_\ell(0), \\
X_\ell &= \bar{f}_{\ell,\mathcal{M}_1^{(\ell)}}, \\
Y_\ell &= \widehat{Var}(f_\ell(t))_{\mathcal{M}_2^{(\ell)}}.
\end{aligned}
\tag{23}
$$

Based on these data, we formulated a test statistic for genetic drift as follows. We grouped filtered lineages into non-overlapping initial frequency bins $b_i$ spanning $(f_{\min}(0), f_{\max}(0))$. In each bin, we computed the sample averages $\langle W_\ell \rangle_{b_i}$, $\langle X_\ell \rangle_{b_i}$, and $\langle Y_\ell \rangle_{b_i}$, excluding those lineages whose variances $Y_\ell$ fell in the lowest and highest deciles of the bin $b_i$. We choose this trimmed mean to conservatively remove outliers that might be driven by strong sequencing noise (e.g. PCR jackpots) or de novo mutations. We estimated standard errors of the average variance in each bin, $\sigma_{Y,b_i}$ from 1000 bootstrap samples of the lineages within each bin. We then defined the test statistic $\hat{\Omega}(\mathcal{L})$ as the inferred slope from bootstrap uncertainty-weighted linear regression of the 'Fano factor' $\langle Y_\ell \rangle_{b_i}/\langle X_\ell \rangle_{b_i}$ against the initial frequency $\log \langle W_\ell \rangle_{b_i}$. Because drift is stronger (and final frequency variation higher) in lineages beginning at smaller initial frequencies, a negative slope indicates the effects of drift.

We compared the observed test-statistic $\hat{\Omega}(\mathcal{L})$ against an empirical null distribution obtained by randomly permuting the initial frequencies across lineages in the same final frequency bin. This procedure conserves both the numbers of lineages and distribution of initial frequencies in each initial frequency bin $b_i$, and should therefore generate a null distribution of test statistics if the variance was independent of initial frequency (as expected in the absence of drift). We recalculated the test statistic $\hat{\Omega}$ for $n = 10^4$ such permutations, $\mathcal{L}_i'$, and estimated a one-sided $p$-value as

$$
p = \frac{1}{n}\sum_i \mathbb{I}(\hat{\Omega}(\mathcal{L}_i') \leq \hat{\Omega}(\mathcal{L})).
\tag{24}
$$

As an example, we applied this approach to $Bc$ lineages falling in a narrow final (day 4) average frequency range in LF/HPP mice (Fig. 5d), as well as for simulated data assuming a similar distribution of lineage fitnesses (Supplementary Fig. 17). For clarity, we plot the variance (rather than the Fano factor) as a function of the initial frequency.

One can invert the regression in Fig. 5d to estimate the effective population size $N_e\tau_e$ in the simple model in Eq. (22). To do so, we define $\alpha = \{\alpha_1, \alpha_2\}$ and $g(w, x)$ as

$$
g_\alpha(w, x) = \alpha_1 \cdot x + \alpha_2 \cdot \frac{x[x/w - 1]}{\log[x/w]},
\tag{25}
$$

and minimize the uncertainty-weighted sum of squared residuals

$$
\alpha^* = \arg\min_\alpha \sum_{b_i} \frac{1}{\sigma_{Y,b_i}^2}[\langle Y \rangle_{b_i} - g_\alpha(\langle W \rangle_{b_i}, \langle X \rangle_{b_i})]^2.
\tag{26}
$$

The bottleneck size inferred from the given collection of lineages $\mathcal{L}$ is then given by $N_e\tau_e = t/\alpha_2^*$, while the strength of sequencing noise is reflected in $D_{\text{eff}} \equiv 1/\alpha_1^*$. We carried out this procedure using the `curve_fit()` function from the SciPy library[49]. We also estimated the uncertainties in these parameters, $\sigma_{N\tau}$ and $\sigma_{D_{\text{eff}}}$, using the diagonals of the uncertainty matrix returned by the `curve_fit()` function. Figure 5e shows the inferred values of $N_e\tau_e$ for the $Bc$ populations across a range of final frequency windows, spanning $(4 \times 10^{-5}, 8.5 \times 10^{-5})$ and $(2.3 \times 10^{-5}, 4 \times 10^{-5})$ in the LF/HPP and HF/HS diets, respectively. Only regressions with relative uncertainty $|\sigma_{N\tau}/\hat{N\tau}| < 1$, and effective sequencing noise uncertainty $|\sigma_{D_{\text{eff}}}/\hat{D}_{\text{eff}}| < 0.25$, are shown.

Further discussion of our algorithm and comparisons to previous approaches for quantifying drift are described in Supplementary Note 7.

## Reporting summary

Further information on research design is available in the Nature Portfolio Reporting Summary linked to this article.

## Data availability

Raw sequencing data from the original Tn-Seq experiments were downloaded from the European Nucleotide Archive, accession no. PRJEB9434, and the raw data from the input libraries were obtained from the authors of the original study [https://doi.org/10.1126/science.aac5992]. Reference genomes for each of the 4 Bacteroides strains were obtained from the National Center for Biotechnology Information (accession no. PRJNA289334). Postprocessed data analyzed in this paper are directly available on Github [https://github.com/bgoodlab/adaptation_tnseq] and in Supplementary Code 1.

## Code availability

All analysis and figure generation code is available on Github [https://github.com/bgoodlab/adaptation_tnseq], as well as in Supplementary Code 1.

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

## Acknowledgements

We thank M. Wu and J. Gordon for providing access to the original data, and S. Walton, Z. Liu, and other members of the Good Lab for useful discussions and feedback on the manuscript. This work was supported in part by the Alfred P. Sloan Foundation grant FG-2021-15708, NIH NIGMS

Grant No. R35GM146949, and a Terman Fellowship from Stanford University. B.H.G. is a Chan Zuckerberg Biohub - San Francisco Investigator.

## Author contributions

Conceptualization: D.P.G.H.W. and B.H.G.; theory and methods development: D.P.G.H.W. and B.H.G.; analysis: D.P.G.H.W. and B.H.G.; writing: D.P.G.H.W. and B.H.G.

## Competing interests

The authors declare no competing interests.
