## [Peer Review File · Nature Communications]

REVIEWER COMMENTS

Reviewer #1 (Remarks to the Author):

This manuscript presents a new analysis of previous published data, where transposon mutant libraries of *Bacteroides* strains were used to colonize several mice with a defined gut microbiota. Here the authors focus on positive selection that occurred in the gut driven by mutations other than the insertion KO's of the library. This is an interesting idea, and the authors make a lot of effort at trying to estimate evolutionary parameters from this sort of data.

The main claims of the paper are that there is positive selection on thousands of previously hidden mutations, that the distribution of fitness effects during short term adaptation differs between species and that there are fitness trade-offs in different diets.

The paper brings a new analysis and some novel conclusions of potential impact to a broad audience, however, in the way it is currently written it is not an easy paper to follow. While some of the key ideas for the analysis are well illustrated in some of the figure panels, I felt that there were quantitative claims that are difficult to extract from the information given in the main text of the manuscript. In addition, the main text does not make clear how the analysis done here differs from previous methods for analyzing data from barcodes.

One major comment that I have regards the statement in the supplementary methods on the filtered data used for the analysis. My doubt here is that the authors calculate frequencies of lineages removing any of the fitness determinants identified by Wu et al. They then use an evolutionary model for the dynamics of the lineages (presumably only the filtered lineages) in which the mean fitness of the population is calculated (eqs 2 in the supplement). I find this not intuitive and I think more justification is needed, as well as possibly the use of simulations to back up the simplifications applied here.

Some other comments:

It is unclear if the ecological environment in which the changes in frequency of the focal lineages are measured is stable or not, across the duration of the experiment.

Line 69: It states, "This suggests that their fitness benefits did not derive from the original Tn insertion, but rather from secondary mutations that accumulated in the library pool prior to colonization." At this point it is unclear to me why these mutations occurred prior to colonization and not during the colonization.

Lines 74-76: Do the authors want to claim that rates of adaptation vary between species and strains or that what varies across strains is the distribution of fitness effects of beneficial mutations?

Line 109: Where does the 10% can from?

Sup Materials

Line 262 what % of reads were excluded?

Line 293, please state the relation of $N_m(t)$ and $\lambda(t)$.

Reviewer #2 (Remarks to the Author):

The authors present a reanalysis of the data generated in an earlier paper by Wu et al. (2015). That study consisted of the introduction of four human gut *Bacteroides* strain transposon mutant libraries, together with 11 other wild-type strains, into gnotobiotic mice, which were given two diets either low fat (LF) or high fat (HF). There was also an alternating diet group but that does not seem to be used in this study. The mice were given the diets for 16 days and fecal samples collected every 4 days.

Transposon libraries are a powerful method to determine genes that are critical for the growth of a microbe in a particular context, by introducing transposons randomly throughout the genome, individual genes are deactivated. Amplicon sequencing of the transposons and a gene specific overhang allows the proportion of transposon mutants with a particular knockout to be estimated. If a particular gene is critical for growth then its knockout will be absent in the follow up sequencing, in Wu et al. they focussed on genes where all mutants with that knockout were consistently associated with a fitness cost in a particular diet. Deducing that these encoded proteins that were important to growth under those conditions. In this study, instead, they focus on those genes, where the knockouts did not have a consistent fitness cost, and particularly those that increase in abundance during the experiment. The premise being that some of these mutants must have acquired mutations elsewhere in their genomes which are responsible for the difference in their fitness to the other knockouts of the same gene.

Major comments:

1) On the whole I felt this was an interesting reanalysis, it certainly made me think, but the limitations inherent in reusing an existing study meant that some of the authors' claims either need to be scaled down or require additional experimental justification. The principle limitation being that they did not directly observe the secondary mutations that they hypothesise are responsible for the fitness effects on the transposon mutants. Take the claim in the abstract:

Abstract line 14: 'we observed positive selection on thousands of previously hidden mutations – most of which were unrelated to their original gene knockouts'. In my opinion, these mutations are still hidden, the authors have inferred they exist but direct evidence is lacking, a more correct statement would be 'we infer the existence of thousands of cryptic mutations', or words to that effect. In my opinion the entirety of claims in the text regarding the concrete nature of the mutations, needs to be adjusted to reflect the reality of the method used here. The alternative would be for the authors to perform some direct validation of these mutations. This need not be too hard as I discuss below.

2) As discussed above, the idea that there are secondary mutations in the transposon libraries, that are responsible for these fitness effects, is central to the conclusions in the paper. However, the actual mechanism by which these mutations are introduced is never considered. I am not an expert in this area of molecular biology although I have enough experience to know that protocols never do exactly what they should. Therefore I do find it plausible that these mutations exist but the authors need to elaborate on this. Where are the mutations coming from, what step in library generation, can differences in the mutation introducing step in the different library preparations, actually explain the differences in fitness distributions between strains and species that are observed. Therefore Line 74: 'This shows that the rates of adaptation in the murine gut can vary between similar gut species, and even between strains of the same species.' may not be correct.

3) Ideally, as mentioned above the authors, would directly validate the existence of these mutations, culturing off the fecal samples would have been performed. I appreciate this is not possible. It would be unethical to repeat the entire original experiment to validate this point. However, repeating the transposon library generation using the protocol in the original paper, and then sequencing individual mutants or the entire library with shotgun metagenomics, would be quite straightforward. The frequency of secondary mutations could then be validated. I know this sounds onerous but if the authors are genuinely proposing transposon libraries as an alternative to neutral barcoding e.g. Line 41 'Here we show that similar evolutionary inferences can be obtained from genome-wide transposon insertion sequencing (Tn-Seq) libraries (22–26), which are routinely employed in functional genomics settings.' then I feel that there is an onus on the authors to understand how the method is working in this context.

Minor comments:

1) Some supplementary figures could be brought into the main text to aid readability of the text.

2) Caption Figure S1: 'the fold change of the top 10 largest lineages at day 16 is plotted against the
???? of other Tn lineages'

Detailed Response to Reviewer Comments

Note: line numbers refer to the track-changes version of the manuscript.

Reviewer #1 (Remarks to the Author):

This manuscript presents a new analysis of previous published data, where transposon mutant libraries of *Bacteroides* strains were used to colonize several mice with a defined gut microbiota. Here the authors focus on positive selection that occurred in the gut driven by mutations other than the insertion KOs of the library. This is an interesting idea, and the authors make a lot of effort at trying to estimate evolutionary parameters from this sort of data. The main claims of the paper are that there is positive selection on thousands of previously hidden mutations, that the distribution of fitness effects during short term adaptation differs between species and that there are fitness trade-offs in different diets.

(1) The paper brings a new analysis and some novel conclusions of potential impact to a broad audience, however, in the way it is currently written it is not an easy paper to follow. While some of the key ideas for the analysis are well illustrated in some of the figure panels, I felt that there were quantitative claims that are difficult to extract from the information given in the main text of the manuscript.

We thank the Reviewer for their overall positive comments. We have made various changes in the main text and figures to make the paper easier to follow, and to better substantiate the quantitative claims made in the main text. We describe some of these in more detail in the specific points below. (See also our response to Comment 4 from Reviewer #2.)

(2) In addition, the main text does not make clear how the analysis done here differs from previous methods for analyzing data from barcodes.

We apologize for the confusion – we have added some additional sentences to the “Contrasting the spectrum...” section [lines 113-119] to better highlight these differences.

To summarize, our analysis differs from previous lineage tracking studies in two major ways:

(i) Since we are starting from a TnSeq library, it was necessary to develop an approach for distinguishing spontaneous mutations (the primary object of our study) from the knockout effects of their original Tn insertions. Based on the Reviewer’s summary above, we are assuming that this particular aspect of our analysis was already sufficiently emphasized in the text.

(ii) The lower read counts and fluctuating environmental conditions in our experiment prevented us from applying existing barcode inference methods, which assume that the fitnesses of the individual mutations are constant and well-sampled across multiple consecutive timepoints. To overcome this issue, developed an alternative approach for inferring selection coefficients from the consistent expansion of shared genetic variants across multiple independent populations

(rather than a single population over time). While simple in principle, this required us to develop methods to ensure that our conclusions were robust to the large amounts of sampling noise generated by the amplicon sequencing protocol. This allows us to make statistically rigorous statements about the collective properties of the adaptive lineages (e.g. their distribution of fitness effects) even when there are large uncertainties in many individual lineages.

In the revised manuscript, we have added some additional text on lines [126-131] in the Main Text to emphasize this second point. We have also added a new panel to Fig. 1 (panels D and G) to better emphasize how our method is able to estimate the distribution of fitnesses of the adaptive lineages.

(3) One major comment that I have regards the statement in the supplementary methods on the filtered data used for the analysis. My doubt here is that the authors calculate frequencies of lineages removing any of the fitness determinants identified by Wu et al. They then use an evolutionary model for the dynamics of the lineages (presumably only the filtered lineages) in which the mean fitness of the population is calculated (eqs 2 in the supplement). I find this not intuitive and I think more justification is needed, as well as possibly the use of simulations to back up the simplifications applied here.

Thanks for bringing this up – this is an important but subtle point. In this case, one can show mathematically that for a well-mixed population (a central assumption in all existing barcode inference methods) it is always possible to reduce the model to the general form in Eq. 2 for arbitrary subsets of the lineages. The main distinction is that the “mean fitness” in this case refers to the mean fitness of the included lineages, rather than the mean fitness of the entire population (which may have been unclear in our previous version). To clarify this point, we have added an additional section to the SI that performs this mathematical derivation and points out the different notions of “mean fitness” [lines 363-385]. We have also added a new panel to Figure 1 (Fig.1B) summarizing the fraction of lineages that were included/excluded from our analysis.

Some other comments:

(4) It is unclear if the ecological environment in which the changes in frequency of the focal lineages are measured is stable or not, across the duration of the experiment.

We apologize for the lack of clarity – yes, the ecological environment that the bacteria encounter is definitely changing over time. Some of these changes are externally imposed (e.g. the alternating diet arms), while others reflect the inherent non-equilibrium dynamics of *in vivo* colonization experiments. Our demonstration that these latter changes also impact the adaptive landscape – even under a constant diet – is one of the major new contributions of our study (Figs. 3 & 4).

To clarify this issue, we have expanded our discussion of the latter point on lines [178-182]. We have also added several new panels to Fig. 1 (panels C-F) illustrating the lineage dynamics in

the alternating diet arms, and added some additional text to the opening of the 2nd and 3rd sections of the Results sections to better set up the problem [113-115; 163-165].

(5) Line 69: It states, “This suggests that their fitness benefits did not derive from the original Tn insertion, but rather from secondary mutations that accumulated in the library pool prior to colonization.” At this point it is unclear to me why these mutations occurred prior to colonization and not during the colonization.

Thanks for pointing this out – we have restructured the text in this section to try to make the overall argument flow more clearly [65-74].

We realized that at this point in the section, understanding *when* the mutations occurred is only of secondary importance (relative to the more surprising claim that they are due to secondary mutations). So we have restructured this paragraph to only focus on the latter point [70-74], and left the discussion of when the mutations arose to Fig 2 [99-105;139-142] where the evidence is presented more explicitly (e.g. Figs. 2A and 2C).

(6) Lines 74-76: Do the authors want to claim that rates of adaptation vary between species and strains or that what varies across strains is the distribution of fitness effects of beneficial mutations?

Thanks for highlighting this issue – this is exactly the distinction that we were trying to draw between Figs. 1 & 2, but we did a poor job explaining this in the original version of the manuscript. To clarify, we believe that our results show that the distribution of fitness effects varies between the *Bacteroides* strains, and this is ultimately what gives rise to the different rates of adaptation in Fig. 1.

In the revised manuscript, we have tried to clarify this issue by adding a new panel to Fig. 2 (panels D and G) to explicitly show our DFE estimates, along with some corresponding discussion in the main text [128-131]. We have also modified the sentence referenced by the Reviewer [80-82] to indicate that it is just an operational definition of the rate of adaptation, which will be contrasted with other metrics later.

(7) Line 109: Where does the 10% come from?

Thanks for pointing this out. We have amended the text [line 128] to include the total number of lineages tracked, so that it will be more clear where the fraction is coming from.

(8) Sup Materials: Line 262 what % of reads were excluded?

Depending on the species and/or sample, approximately ~2-10% of reads were excluded because they mapped to multiple locations.

(9) Line 293, please state the relation of $N_m(t)$ and $\lambda(t)$.

Thank you for pointing out this typo – the $N_m(t)$ and $\lambda(t)$ variables were actually left over from an earlier draft, so we have changed this to reflect the $\Lambda_m(t)$ notation we used in the rest of this work.

The $\Lambda_m(t)$ parameter represents the effective rate of genetic drift. In the simplest models, $1/\Lambda_m(t)$ is equal to the product of the effective population size and the effective generation time (see [846-850] in SI Section 8 “*Estimating the strength of genetic drift in vivo*”). We chose the more general nomenclature $\Lambda_m(t)$ to capture scenarios where the strength of genetic drift is not necessarily captured by the census population size (see Ghosh and Good, PNAS 2022). We have added a reference to this work when Eq. 2 is first introduced.

Reviewer #2 (Remarks to the Author):

The authors present a reanalysis of the data generated in an earlier paper by Wu et al. (2015). That study consisted of the introduction of four human gut *Bacteroides* strain transposon mutant libraries, together with 11 other wild-type strains, into gnotobiotic mice, which were given two diets either low fat (LF) or high fat (HF). There was also an alternating diet group but that does not seem to be used in this study.

A quick point of clarification – we do use the alternating diet groups in our study. These mice were used to obtain the purple points in Fig. 3C, and make up a sizeable fraction of the “microenvironments” in Fig. 4. (On a more technical level, we also used them to increase the sample size for the day 0-4 measurements, since the fixed diet and alternating diet arms are identical in this time interval, see [670-672] in SI Section 6).

However, we agree that this point was not sufficiently emphasized in the first half of the paper and could easily be missed. To eliminate potential confusion, we have added several new panels in Figure 1 to highlight the lineage dynamics in mice from the alternating diet groups (HLH and LHL), which should help signal that we are also using these data in our study.

The mice were given the diets for 16 days and fecal samples collected every 4 days. Transposon libraries are a powerful method to determine genes that are critical for the growth of a microbe in a particular context, by introducing transposons randomly throughout the genome, individual genes are deactivated. Amplicon sequencing of the transposons and a gene specific overhang allows the proportion of transposon mutants with a particular knockout to be estimated. If a particular gene is critical for growth then its knockout will be absent in the follow up sequencing, in Wu et al. they focused on genes where all mutants with that knockout were consistently associated with a fitness cost in a particular diet. Deducing that these encoded proteins that were important to growth under those conditions. In this study, instead, they focus on those genes, where the knockouts did not have a consistent fitness cost, and particularly those that increase in abundance during the experiment. The premise being that some of these mutants must have acquired mutations elsewhere in their genomes which are responsible for the difference in their fitness to the other knockouts of the same gene.

This is an excellent summary of our central premise.

Major comments:

(1) On the whole I felt this was an interesting reanalysis, it certainly made me think, but the limitations inherent in reusing an existing study meant that some of the authors' claims either need to be scaled down or require additional experimental justification. The principle limitation being that they did not directly observe the secondary mutations that they hypothesize are responsible for the fitness effects on the transposon mutants.

We thank the reviewer for their constructive comments. We agree that it is important to acknowledge the limitations of our study, and to clearly articulate what we can (or cannot) rigorously show. We have made various revisions to the text to address the specific comments below:

Take the claim in the abstract:

Abstract line 14: 'we observed positive selection on thousands of previously hidden mutations – most of which were unrelated to their original gene knockouts'. In my opinion, these mutations are still hidden, the authors have inferred they exist but direct evidence is lacking, a more correct statement would be 'we infer the existence of thousands of cryptic mutations', or words to that effect. In my opinion the entirety of claims in the text regarding the concrete nature of the mutations, needs to be adjusted to reflect the reality of the method used here. The alternative would be for the authors to perform some direct validation of these mutations. This need not be too hard as I discuss below.

We appreciate the Reviewer's point here. On the one hand, we'd like to emphasize that our modeling framework does provide rigorous *statistical* evidence that mutations (or other forms of heritable phenotypic variation) must be responsible for the effects we observe – to our knowledge, this is the only mechanism that could explain the consistent expansions we observe across replicate mice in distinct time intervals. This reliance on mathematical modeling is similar to previous genetic barcoding experiments (Levy et al, *Nature* 2015; Nguyen Ba et al, *Nature* 2019; Jasinska et al, *Nature Eco Evo* 2020; Karlsson et al, *Nature* 2023), which also did not directly observe the mutations driving individual barcodes. Similar to these works, our goal is to show that one can obtain useful information about the evolutionary consequences of these mutations even without knowing their precise identities.

That being said, we agree with the Reviewer's point that the identities of the mutations are still hidden (this is an important point that we emphasized in the Discussion). We have therefore taken the Reviewer's suggestion and amended the abstract [14-18], as well as various other locations in the main text [72-74, 90, 112, 223, 230, 232, 258-267], to better reflect that we can only infer the existence – and not the precise identities – of the mutations studied in this work. As part of these revisions, we have also made an effort to limit our use of the term "mutation" to

scenarios where the inferential nature of the evidence is clear, opting for more neutral terms like “variant” or “lineage” whenever possible. Together, these changes should help clarify the nature and limitations of the statistical inference approach that we’ve utilized here.

Finally, we have made a major revision to Fig. 4, which shows how the fitness tradeoffs across environments can shed light on the phenotypic nature of the underlying mutations, even without directly sequencing them. For example, we show that the tradeoff signatures some lineages closely resemble loss-of-function mutations in particular pathways, while others are distinct from any gene-level knockouts. We believe that this analysis is an important new contribution of our work, which goes a step toward addressing the Reviewer’s questions regarding the nature of the mutations involved. Since the Reviewer did not comment on this part of our analysis, we inferred that we did a poor job explaining it in our initial draft. We have therefore revised the figure to make it easier to read, including a schematic illustration of the overall premise. We have also added some additional examples of non-knockout-like lineages in other species in Supplemental Fig. S15.

(2) As discussed above, the idea that there are secondary mutations in the transposon libraries, that are responsible for these fitness effects, is central to the conclusions in the paper. However, the actual mechanism by which these mutations are introduced is never considered. I am not an expert in this area of molecular biology although I have enough experience to know that protocols never do exactly what they should. Therefore I do find it plausible that these mutations exist but the authors need to elaborate on this. Where are the mutations coming from, what step in library generation, can differences in the mutation introducing step in the different library preparations, actually explain the differences in fitness distributions between strains and species that are observed. Therefore Line 74: 'This shows that the rates of adaptation in the murine gut can vary between similar gut species, and even between strains of the same species.' may not be correct.

Thanks for bringing this up – this comment made us realize that our initial draft had not sufficiently emphasized the fact that secondary mutations are also commonly observed in more traditional “neutral” barcoding systems (Levy et al, *Nature* 2015; Nguyen Ba et al, *Nature* 2019; Jasinska et al, *Nature Eco Evo* 2020; Vasquez et al, *Cell Host & Microbe* 2021, Karlsson et al, *Nature* 2023). The precise causes of these mutations are not known in most cases, but they are thought to arise from a mixture of outgrowth during library creation and/or other mutagenic aspects of the library creation protocol. Our finding that similar mutations are also present in TnSeq libraries is consistent with this previous literature – our new contribution is to show that many of these mutations seem to provide a fitness benefit *in vivo*.

At present, it is not clear whether the higher rates we observe here are driven by specific features of the TnSeq protocol used in the original study (e.g. antibiotic selection or aerobic conjugation) or the increased importance of non-point-mutation processes (e.g. phase variation) that are known to occur at high rates in some *Bacteroides* species (Jiang et al, *Science* 2019; Lan et al, *Science Adv* 2023; Channin et al, *bioRxiv* 2023). Regardless, we believe that the phenotypic diversity revealed by Figs. 3 and 4 shows that these standing variants are not all

caused by a single high-rate mutation (e.g. the self-diploidization mutations in Venkataram et al, *Cell* 2016), but instead comprise a broader adaptive landscape that can drive *in vivo* evolution.

To clarify these points, we have added some additional discussion along these lines when secondary mutations are first mentioned in the Results [99-105], as well as an additional paragraph in the Discussion [258-267]. This should provide better context for the secondary mutations hypothesis, and its relationship to previous observations in traditional “neutral” barcoding systems.

Finally, we appreciate the Reviewer’s point that the differences between species on Line 74 could also be caused by differences that occur during library creation. While we can verify that the libraries for the different species examined in this study were created using the same protocol (p. 3-5 of the Supplemental Information of Wu et al, *Science* 2015), we cannot exclude the possibility that the species responded in different ways to the same protocol conditions. We have therefore revised the sentence on Line 74 [now 80-82] to better reflect the operational nature of our observation:

“This shows that the apparent rates of adaptation – as measured by the expansion of the largest TnSeq lineages – can vary between closely-related commensal gut species, and even between strains of the same species.”

This also helps address a related comment by Reviewer 1 above (Comment #6).

(3) Ideally, as mentioned above the authors, would directly validate the existence of these mutations, culturing off the fecal samples would have been performed. I appreciate this is not possible. It would be unethical to repeat the entire original experiment to validate this point. However, repeating the transposon library generation using the protocol in the original paper, and then sequencing individual mutants or the entire library with shotgun metagenomics, would be quite straightforward. The frequency of secondary mutations could then be validated.

I know this sounds onerous but if the authors are genuinely proposing transposon libraries as an alternative to neutral barcoding e.g. Line 41 'Here we show that similar evolutionary inferences can be obtained from genome-wide transposon insertion sequencing (Tn-Seq) libraries (22–26), which are routinely employed in functional genomics settings.' then I feel that there is an onus on the authors to understand how the method is working in this context.

We agree that understanding the molecular basis of the mutations we have inferred would be highly valuable. However, we believe that this is best left for a future study, for the reasons described below:

(i) Our response to the previous point makes it clear that the existence of secondary mutations alone is not surprising, since they have also been observed in existing neutral barcoding systems. Understanding the mutagenic aspects of different barcoding methods would be a valuable contribution, but any comparisons to existing neutral barcoding systems would require

us to carry out similar experiments using those systems as well – this is a much larger undertaking that would be beyond the scope of our present work.

(ii) The more surprising part of our analysis is that so many of these pre-existing mutations seem to provide a fitness benefit in the complex environment of the mouse gut. Demonstrating that a particular secondary mutation provides a fitness benefit would require us to repeat the entire experiment in a new set of mice – we agree with the Reviewer that this would be unethical for the reasons they noted above.

(iii) In collaboration with another group, we are currently working on a follow-up study (also in *B. theta*, but using a different barcoding method) in which we were able to culture and sequence a large number of isolates from mouse fecal samples over time. An extension of the method we developed here revealed a number of pre-existing beneficial mutations in this new system as well, and the isolate sequencing allowed us to validate these mutations in a handful of cases. The identified mutations consisted of a mixture of point mutations and alterations of phase variable loci, which is consistent with our discussion above. While these unpublished results cannot be incorporated into our present study, we hope that they help illustrate that our inferences are not implausible, and that extensions of our modeling approach can be useful in other contexts.

In the revised manuscript, we have added an additional section to the SI ([603-615] and Supplemental Figure S13) that shows how our fitness inferences can be used to identify timepoints where future isolate sequencing efforts would be most informative. This highlights what we believe is the central contribution of our study: the development of a modeling framework for inferring the spectrum of beneficial mutations in these (and other) barcoding systems in complex *in vivo* settings, where the simplifying assumptions of previous lineage tracking approaches do not necessarily hold. Together with the revisions discussed in Points #1 and #2 above, we hope that this addresses the Reviewer's concerns about the nature of the mutations we have inferred.

Minor comments:

(4) Some supplementary figures could be brought into the main text to aid readability of the text.

Thanks for the suggestion (Reviewer 1 had a similar comment above). We have added several new panels to the main Figures to improve the readability of the text:

(i) We have added a new panel to Figure 1 (panel B) summarizing the total number of Tn lineages in each library and the number retained for our analysis.

(ii) We have added new subpanels to Fig. 1C-F to illustrate the lineage dynamics under the fluctuating dietary conditions.

(iii) We have added another new panel to Figure 1 (panel G) showing Tn lineage trajectories for

three example genes that illustrate the central premise of our study. One of the genes shows consistent expansion across many Tn lineages, consistent with a traditional beneficial gene knockout. The other two genes highlight examples of adaptive Tn lineages, which deviate from the other lineages in the same gene.

(iv) We have added new panels to Figure 2 (panels D and G) which illustrate how the rank order curves in panels C and F can be used to infer the fitness spectrum of the adaptive lineages.

(v) As described in our response to the Reviewer's point #1 above, we have added a new schematic panel to Figure 4 to illustrate the basic idea of the approach that was previously described in the SI.

(vi) Finally, we have created added a new figure to the Main Text (Figure 5) that summarizes our results on *de novo* mutations (previously SI Figs. S15 & S16) and inferring the rate of genetic drift *in vivo* (previously SI Fig. S17). We believe that this latter example represents a significant advance over existing methods like STAMPR (Abel et al, *Nature Methods* 2015; Campbell et al *Nature Comms* 2023), which do not account for the widespread selection that we have observed.

Note that while we were revising these supplemental figures for the broader audience of the Main Text, we discovered a simpler version of our approach to infer the rates of genetic drift which is more amenable to direct visualization (Fig. 5D). We have therefore also updated the corresponding section of the methods [852-927] to reflect this simpler procedure.

5) Caption Figure S1: 'the fold change of the top 10 largest lineages at day 16 is plotted against the ???? of other Tn lineages'

Thanks for pointing this out – we have fixed this typo in the revised manuscript.

REVIEWERS' COMMENTS

Reviewer #1 (Remarks to the Author):

The authors have made several changes in the main manuscript, in the supplementary information and main figures which address the issues I raised in my previous revision.

I think the manuscript is now easier to follow and I thank the authors for now clarifying the analysis regarding the filtered data.

Reviewer #2 (Remarks to the Author):

I appreciate the effort the authors have put into responding to my queries about the nature and origin of these mutations. I genuinely found their response helpful in aiding my understanding of these types of experiments and am happy that the revised text addresses my original concerns. I look forward to reading their follow up study when that is published.